# The frequency response of networks as open systems

Amirhossein Nazerian [1], Malbor Asllani [2], Melvyn Tyloo [3,4], Wai Lim Ku [5,6,7] & Francesco Sorrentino [1,8] ✉

Many biological, technological, and social systems can be effectively described as networks of interacting subsystems. Typically, these networks are not isolated objects, but interact with their environment through both signals and information that are received by specific nodes with an input function or released to the environment by other nodes with an output function. An important question is whether the structure of different networks, together with the particular selection of input and output nodes, is such that it favors the passing or blocking of such signals. For a given network and a given choice of the input and output nodes, the $\mathcal{H}_2$-norm provides a natural and general quantification of the extent to which input signals–whether deterministic or stochastic, periodic or arbitrary–are amplified. We analyze a diverse set of empirical networks and find that many naturally occurring systems, such as food webs, signaling pathways, and gene regulatory circuits, are structurally organized to enhance the passing of signals; in contrast, the structure of engineered systems like power grids appears to be intentionally designed to suppress signal propagation.

Networked dynamical systems are omnipresent in nature and engineering, ranging from neuronal dynamics to electric power grids. They consist of many interacting units, each with its own intrinsic parameters and degrees of freedom, whose mutual coupling typically gives rise to rich collective behaviors. Usually, these networks are not isolated from their environment, but are subjected to external perturbations, operational changes, and external stimuli. Examples are neuron networks subject to sensory stimuli[1,2], power grids that both absorb power produced at the generators and deliver power at the loads[3–5], and networks subject to the effects of environmental noise, such as ecological networks[6]. At the same time, networks produce outputs that affect the environment. Hence, the response of networked systems to external inputs essentially depends on the choice of the input nodes and output nodes and on the structure and characteristics of the network connectivity.

While a large body of literature has studied control strategies for networks, see e.g.,[7–10], this paper focuses on the response of networked systems to various environmental inputs, not limited to control signals. References[11–16] have studied the response of a networked system to external stimuli. These studies have primarily focused on graphs with symmetric connectivity or, at most, on asymmetric but normal adjacency matrices. Yet, most real-world networks are highly non-normal, strongly directed, and often exhibit pronounced hierarchical organization—structural features that profoundly shape their dynamical response[17–25]. In such networks, non-normality acts as an algebraic expression of directedness and emerges from strong asymmetries in the adjacency pattern with a clear source and sink structure, and induces strongly biased directions of flow. Structurally, highly non-normal networks are close to acyclic, with a dominant directed acyclic backbone and only sparse or weak cycles; this can be

---

[1]Department of Mechanical Engineering, University of New Mexico, Albuquerque, NM, USA. [2]Department of Mathematics, Florida State University, Tallahassee, FL, USA. [3]Living Systems Institute, University of Exeter, Exeter, UK. [4]Department of Mathematics and Statistics, Faculty of Environment, Science, and Economy, University of Exeter, Exeter, UK. [5]Center for Applied Data Science and Analytics, Howard University, Washington, DC, USA. [6]Center for Sickle Cell Disease, Howard University, Washington, DC, USA. [7]Department of Medicine, Howard University, Washington, DC, USA. [8]Max Planck Institute for the Physics of Complex Systems, Dresden, Germany. ✉e-mail: fsorrent@unm.edu

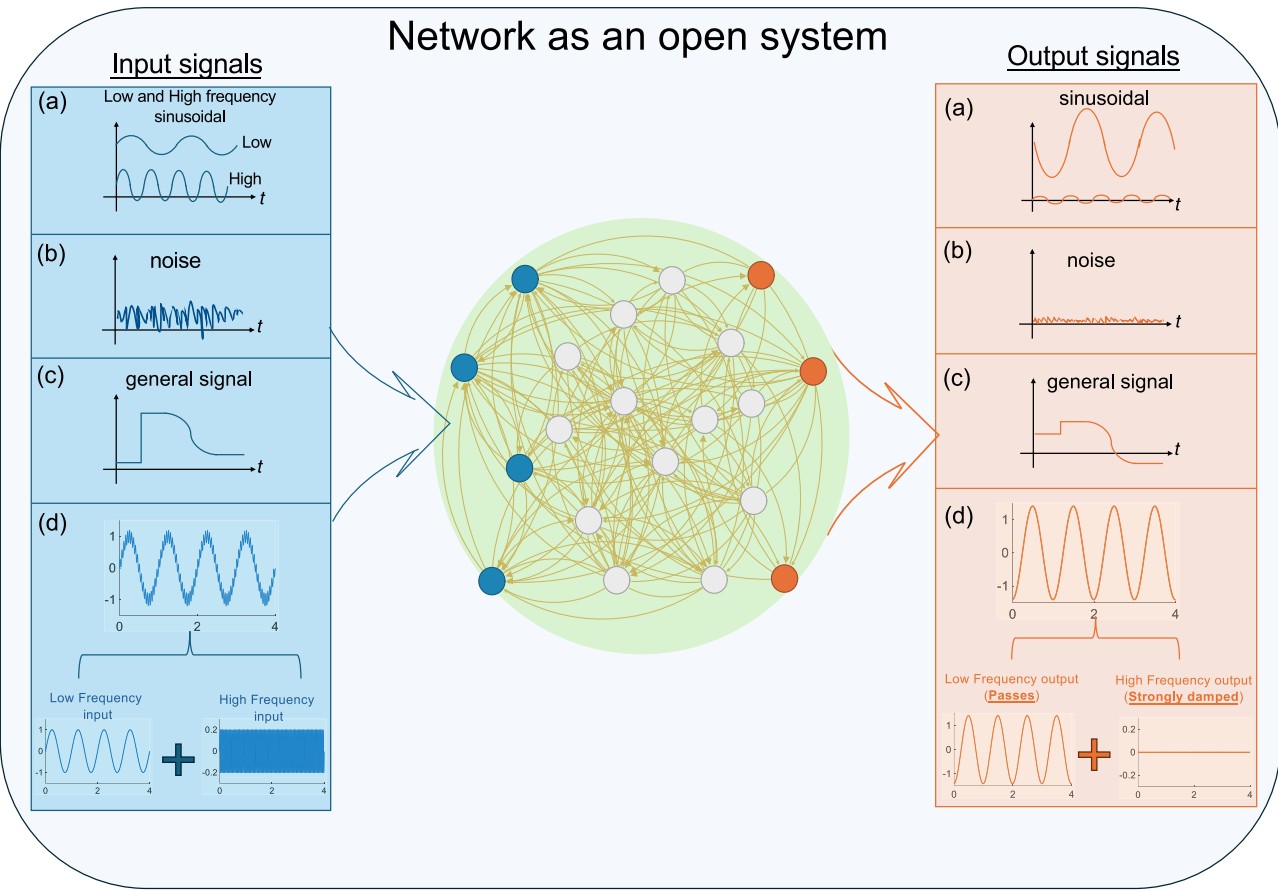

**Fig. 1 | Networks as open systems.** The schematic summarizes the goal of the paper in studying how an external signal is altered when it passes through a network. The network is subjected to various input signals from the environment, which are only received at the input nodes (in blue.) The response of the network is measured at some output nodes (in orange). Several input signals that are considered in this paper and the corresponding output signals are shown. In (**a**), a low-frequency sinusoidal input signal may experience a phase shift and amplification, while a high-frequency one may be damped. The case in (**b**) is that of noise suppression. In (**c**), the shape of a general output signal may not be the same as its corresponding input signal. In (**d**), the input signal contains both a low frequency and a high frequency components. In the output signal, the low frequency passes while the high frequency is strongly damped and thus effectively blocked.

quantified, for example, by measuring the distance from the largest directed acyclic subgraph, a quantity that indicates how close the network is to a loopless structure[17–19].

Consistent with this view, recent work shows that increasing directedness and non-normality leads to flows that are more localized, strongly biased downstream, and effectively irreversible, so that these architectures naturally favor oneway signal transfer toward terminal components and sink nodes[22,25]. Motivated by these properties, our approach does not assume symmetry and addresses the response to general input signals, such as sinusoidal, periodic, and stochastic, within a unified framework. Recent works have focused on how different nonlinearities affect the propagation patterns and stability in complex networks[26,27]. Here, instead, we aim at characterizing whether specific types of networks tend to facilitate or block signal transmission, in terms of the combined effect of network structure as well as the corresponding dynamics.

This paper examines how complex networks of coupled dynamical systems respond to environmental interactions, where external signals are received at designated input nodes and measured at specific output nodes[28–31]. An open network is defined by (i) its internal topology, i.e., the set $\mathcal{N}$ of the network nodes and the set $\mathcal{E}$ of directed, possibly weighted, edges that connect them, (ii) the set of input nodes $\mathcal{I} \subseteq \mathcal{N}$ where external stimuli are applied, and (iii) the set of output nodes $\mathcal{O} \subseteq \mathcal{N}$ where the response is measured. As illustrated schematically in Fig. 1, the network structure and its internal connectivity shape how input signals-including sinusoidal, periodic, and noisy-are

transformed as they propagate from input to output, with different frequency components affected by amplification, attenuation, or distortion, depending on the structural properties of the network. Such signals are ubiquitous in natural and technological systems-for example, gene regulatory networks responding to circadian rhythms[32], power grids operating at standardized frequencies of 50–60 Hz[33], and neuronal circuits processing signals across a broad frequency range of roughly 0.1–100 Hz[34]-and we emphasize that our framework also applies to noise and arbitrary signals.

In panel (a) of Fig. 1, a low-frequency sinusoidal input may undergo phase shift and amplification, whereas a high-frequency input is attenuated. Panel (b) illustrates a network that suppresses noise. In panel (c), the output signal generally differs in shape from its corresponding input. In panel (d), the input contains both low- and high-frequency components; in the output, the low-frequency component passes through, while the high-frequency component is strongly attenuated and effectively filtered out. The Methods Section "Frequency response example" presents a numerical example of signal transformation when it passes through a real connectome network.

In this paper, we adopt a frequency-response viewpoint and quantify signal amplification using the $\mathcal{H}_2$-norm, which captures responses to sinusoidal, periodic, noisy, and more general inputs. Our analysis investigates how the topology of a network and the placement of its input and output nodes shape the amplification or attenuation of external signals. We establish a direct relationship between the network's topology, its transfer function, and the controllability Gramian,

which serves as a central tool in our study. This framework allows us to assess whether empirical networks tend to amplify (*pass*) or suppress (*block*) external inputs for given input-output configurations.

Building on this framework, we apply it to a broad collection of empirical networks from biology, technology, and social systems in order to uncover structural principles governing how they transform signals from input to output. Key findings are that many biological networks-such as food webs, signaling pathways, and gene regulatory networks-tend to exhibit architectures that enhance the passing of signals, supporting the efficient flow of biomass, information, or regulatory activity. This signal-passing behavior is especially evident in directed acyclic graphs (DAGs), namely directed networks that contain no directed cycles, which are known to closely approximate many natural real-world networks[18] and suggest that such architectures inherently promote signal propagation. Motivated by this, we analytically compute the DC gain, i.e., the input-to-output gain when the input is a constant signal for DAG networks, thereby linking amplification or attenuation directly to the number and length of the input-output pathways. In contrast, engineered systems such as power grids-better described as undirected (symmetric) graphs-appear to be purposefully designed to suppress signal amplification, reflecting the need for regulation of transmitted signals, such as voltage phase differences, to ensure stable and synchronized operation.

## Results

### Frequency analysis of networks as open systems

Many natural and engineered systems-such as gene networks, power grids, and neural circuits-can be modeled as open networks, where each node evolves through intrinsic dynamics, interacts with neighbors, and communicates with the environment via inputs and outputs. While inherently nonlinear, such systems can often be approximated near equilibrium by a linear time-invariant (LTI) model, enabling powerful analysis through linear control theory.

Throughout this work we consider a general nonlinear model for the network dynamics of the form, $\dot{z}_i(t) = f(z_i(t)) + \sigma \sum_j \tilde{A}_{ij} g(z_i(t), z_j(t)), i = 1, \ldots, N$, where $z_i(t) \in \mathbb{R}$ is the state of system $i$, the function $f : \mathbb{R} \to \mathbb{R}$ describes the intrinsic dynamics of each node, $g : \mathbb{R} \times \mathbb{R} \to \mathbb{R}$ the pairwise interactions, the scalar $\sigma$ is the global coupling strength, and $\tilde{A} \in \mathbb{R}^{N \times N}$ is the directed adjacency matrix of the network. With suitable choices of the functions $f$ and $g$, this general framework has been applied to model diverse networked systems, such as power grids, food webs with competitive and mutualistic interactions, gene-regulatory networks, and neuronal population dynamics. Since many real-world networked systems operate near a stable equilibrium, we study here the linearized dynamics around such an equilibrium, $\dot{x}_i = -A_{ii}x_i + \sum_j A_{ij}x_j, i = 1, \ldots, N$, where the interdependencies among the linearized states $x_i$ of the network nodes are captured by the network Jacobian matrix $A = [A_{ij}]$. Since here we consider a stable fixed point, all eigenvalues of the Jacobian matrix $A$ have a negative real part. For more details on the steps involved in the linearization, see the Methods Section "From nonlinear to linearized equations". We remark that the network Jacobian $A$ shares a similar structure as the network adjacency matrix $\tilde{A}$, which is demonstrated with an example in the Methods Section "From nonlinear to linearized equations".

This formulation leads to a networked system governed by the standard LTI equations:

$$\begin{aligned} \dot{\boldsymbol{x}}(t) &= A\boldsymbol{x}(t) + B\boldsymbol{u}(t), \\ \boldsymbol{y}(t) &= C\boldsymbol{x}(t), \end{aligned} \tag{1}$$

where $\boldsymbol{x}(t) \in \mathbb{R}^N$ is the network state vector, and $\boldsymbol{u}(t) \in \mathbb{R}^m$ is the vector of external input signals acting on the network, reflecting its open nature. The output vector $\boldsymbol{y}(t) \in \mathbb{R}^p$ collects signals from selected nodes. The matrix $A$ is the network Jacobian introduced before, the

matrices $B$ and $C$ encode information on the nodes where inputs enter the network and the nodes from which outputs are measured, respectively. Each column of $B$ corresponds to one input signal and contains a single 1. If $B_{ij} = 1$, this means that input $j$ is received at node $i$. Similarly, each row of $C$ corresponds to one output signal and also contains a single 1. If $C_{ij} = 1$, this means that output $i$ is measured at node $j$. In short, each input is injected at exactly one node, and each output is read from exactly one node.

The LTI approximation provides a principled and analytically tractable framework for studying networked systems operating near stable fixed points, during short transients, or under weak perturbations, where the linearized dynamics captures the dominant system behavior. This includes many biological and engineered systems functioning under steady-state conditions. Also, systems near criticality, such as second-order phase transitions, where linear responses capture essential behavior[35], can be modeled as LTI systems. Even in synchronization problems, governing models in neurodynamics, the LTI framework remains valid when the system is close to the onset of collective oscillations, which typically arise through a Hopf bifurcation occurring at a specific mode (e.g., zero wavenumber or the zero coupling eigenvalue)[36–38]. In such cases, the dynamics can be linearized around the fixed point prior to bifurcation, yielding time-invariant Jacobians that characterize local behavior. Crucially, we find an important relation with the non-normality of the network Jacobian matrix, which is known to induce transient amplification and shape control energy[39] and is an ubiquitous feature of real-world networks. Finally, one should note that many-body interactions can also be accounted for within our framework. Indeed, when linearizing the dynamics around a stable fixed point, the interaction can, at leading order, be described by an effective pair-wise directed network through the Jacobian. In what follows, we restrict our attention to this first-order (linear) approximation, although higher-order terms, e.g., in phase-reduction expansions, can generate explicit many-body effective interactions beyond this description[40].

In this work, we adopt a frequency-domain approach to characterize the dynamics of networked open systems. Any periodic input can be decomposed into a superposition of sinusoids via Fourier analysis, enabling the study of the system's frequency response to generic inputs. For the LTI model in Eq. (1), the transfer function[41] is defined as

$$G(s) = C(s\mathbb{I} - A)^{-1}B. \tag{2}$$

By evaluating the transfer function at $s = J\omega$, where $\omega$ is the input frequency and $J = \sqrt{-1}$, we probe the system's steady-state response to sinusoidal inputs. After the transient has elapsed, the output oscillates at the same frequency $\omega$ of the input signal, but with altered magnitude and phase, given by

$$\text{Magnitude} = 20\log_{10}|G(J\omega)|, \quad \text{Phase} = \angle G(J\omega). \tag{3}$$

Specifically, if the input signal $u(t) = \sin(\omega t)$, then the output signal $y(t) = \text{Magnitude} \cdot \sin(\omega t + \text{Phase})$[41]. The factor $20\log_{10}(\cdot)$ in Eq. (3) expresses the magnitude $|G(J\omega)|$ in decibels (dB), the conventional logarithmic unit for system gain. This frequency-based formulation underpins our analysis and enables principled insights into how the network structure shapes signal transmission and network performance.

An important case of the frequency response analysis arises in the low-frequency limit, which corresponds to the case of slowly varying or constant inputs, such as step functions[42]. This steady-state behavior is captured by the so-called DC gain. In this limit, the frequency variable approaches zero, $\omega \to 0$. We have provided a general formulation for the DC gain in the Methods Section "DC gain".

## $\mathcal{H}_2$-Norm

In this work, the $\mathcal{H}_2$-norm[43] serves as our principal analytical tool to quantify how the structure of a networked system shapes the amplification of inputs into outputs. For a given open network characterized by the triplet $A$, $B$, $C$, the $\mathcal{H}_2$-norm provides a universal and versatile measure of input-to-output amplification with general relevance in at least three fundamental scenarios: arbitrary time-varying inputs, sinusoidal inputs across the frequency spectrum, and noisy (stochastic) inputs. This classical optimal control performance index measures the total energy transfer from inputs to outputs[43,44], quantifies variance amplification in stochastically driven linear systems[45], and coincides with the trace of the Gramian in linear time-invariant settings[43,44]. Intuitively, it reflects both the energy of the impulse response[46] and the sensitivity of the system's output to random input fluctuations.

The $\mathcal{H}_2$-norm squared for the system with transfer function (2) measures the energy of the response to a unit impulse and can be computed as

$$
\begin{aligned}
\| G \|_2^2 &:= \frac{1}{2\pi} \operatorname{Tr}\left[ \int_{-\infty}^{+\infty} G(J\omega)^H G(J\omega) d\omega \right] \\
&= \operatorname{Tr}\left[ \int_0^{+\infty} C e^{At} BB^\top e^{A^\top t} C^\top dt \right] = \operatorname{Tr}[C W_c C^\top],
\end{aligned}
\tag{4}
$$

where $W_c$ is the continuous-time controllability Gramian[47] and is defined as

$$
W_c = \int_0^{+\infty} e^{At} BB^\top e^{A^\top t} dt,
\tag{5}
$$

which, for simplicity, we refer to as the Gramian throughout the paper. An important observation from Eq. (4) is that the $\mathcal{H}_2$-norm squared provides a measure of amplification to all frequencies between 0 and $\infty$, as well as to impulse time signals.

The matrix $W_c$ in (5) is symmetric and positive semi-definite and is finite if and only if the Jacobian matrix $A$ has all eigenvalues with negative real parts. The matrix $W_c$ can be easily calculated by solving the following continuous-time Lyapunov equation[47] for $W_c$:

$$
A W_c + W_c A^\top + BB^\top = 0.
\tag{6}
$$

For more information on the properties of the Controllability Gramian (5), see the Methods Section "Properties of the controllability Gramian", which in particular discusses the important modularity property of the trace of the output Gramian $\operatorname{Tr}(W_c^{out})$. Namely, for any choice of the sets of input nodes and output nodes, the trace $\operatorname{Tr}(W_c^{out})$ can be written as a sum of individual contributions, each one of which corresponds to a pair of input and output nodes. This has direct and important consequences, as it implies that if we assign the set of output nodes, a set of $k$ input nodes that minimizes (maximizes) $\operatorname{Tr}(W_c^{out})$ is simply composed of the $k$ nodes that provide the largest (smallest) individual contributions. An analogous conclusion is obtained in the case that a given set of input nodes is assigned, and one wants to select an optimal set of $k$ output nodes. Furthermore, Supplementary Note 1 relates the output standard deviation to the $\mathcal{H}_2$-norm via the Gramian spectrum, and Supplementary Note 2 provides upper bounds for the Bode integral in terms of the eigenvalues and eigenvectors of the matrix $A$.

Up to this point, we have examined the overall network amplification of input signals across frequencies, as quantified by the $\mathcal{H}_2$-norm. However, it is possible that a particular network will pass certain frequencies and block others, which can be studied by using Bode plots[48]. To illustrate the network response from a local, analytically tractable perspective, we next introduce a simple solvable model

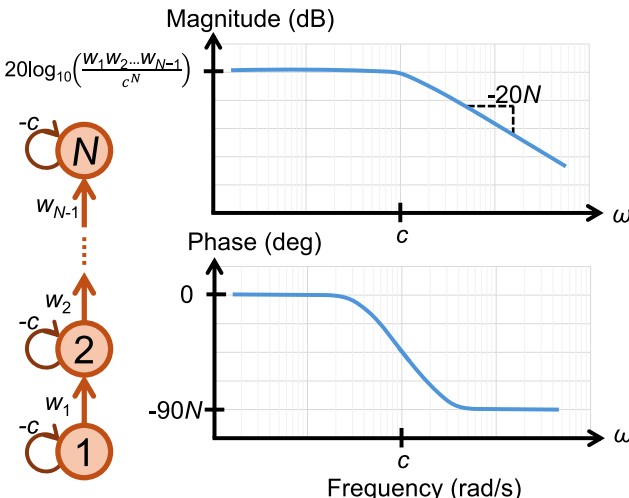

**Fig. 2 | Bode diagram for chain network.** Panel a shows the unidirectional chain network with $N$ nodes and positive link weights $w_1$, $w_2$, ..., $w_{N-1}$ and the self-loops with the weights $-c \leq 0$. Panel b shows the Bode plot corresponding to this system when node 1 is the input node and node $N$ is the output node. In both plots, the $\omega$-axis is on a log scale.

that clarifies how individual frequencies are processed and will serve as a reference point for our later analysis.

**Example 1.** Any periodic input signal can be decomposed into different frequencies through its Fourier series decomposition, which allows the application of frequency response analysis to generic periodic input signals. We focus on the unidirectional chain network with $N$ nodes shown in Fig. 2, where we assume that node 1 is the input node and node $N$ is the output node. For this example, the network Jacobian $A$ can be written as follows,

$$
A = \begin{bmatrix}
-c & & & & \\
w_1 & -c & & & \\
& w_2 & -c & & \\
& & \ddots & \ddots & \\
& & & w_{N-1} & -c
\end{bmatrix},
\tag{7}
$$

where $w_i$ represents the coupling from node $i$ to node $i + 1$, $i = 1, \ldots, N - 1$ and $-c \leq 0$ is the stabilizing self-feedback coefficient at each individual node. The transfer function for this network is,

$$
G(s) = \frac{w_1 w_2 \ldots w_{N-1}}{(s+c)^N} = \frac{\xi_N}{(s+c)^N},
\tag{8}
$$

where $\xi_N = w_1 w_2 \ldots w_{N-1}$ is the product of all the node-to-node couplings along the chain with $N$ nodes. The schematics for the Bode plot of this system is shown in Fig. 2 that summarizes our results, where we sketch the Bode plot based on the parameters $c, N, w_1, \ldots, w_{N-1}$. We see that for frequencies $\omega$ below $c$, the rate of amplification is given by the DC gain, but as $\omega$ grows larger than $c$, the amplitude of the output signal decreases with a rate $\propto 1/\omega^N$. The phase plot has a peak and a low plateau corresponding to low and high frequencies, respectively. A negative phase shift arises in the output signal compared to the input signal, by as much as $-90N$ degrees in large frequencies. The cornering frequency is the frequency at which the power output of a system is reduced to half of its passband value. This corresponds to a 3 dB drop in the magnitude of the signal. Figure 2 shows that the cornering frequency is equal to $c$.

**Table 1 | The coefficient $\phi_N$ in Eq. (9) for different values of the system order $N$**

| $N$ | 1 | 2 | 3 | 4 | 5 | 6 |
|-----|---|---|---|---|---|---|
| $\phi_N$ | $\frac{1}{2}$ | $\frac{1}{4}$ | $\frac{3}{16}$ | $\frac{5}{32}$ | $\frac{35}{256}$ | $\frac{63}{512}$ |

The $\mathcal{H}_2$-norm squared for $G$ is

$$\| G \|_2^2 = \frac{\binom{2N-2}{N-1}}{2^{2N-1}} \frac{\xi_N^2}{c^{2N-1}} = \phi_N \frac{\xi_N^2}{c^{2N-1}}. \tag{9}$$

The details of the calculation can be found in the Supplementary Material.

Table 1 displays $\phi_N$ for $N = 1, 2, ..., 6$, from which we see that the coefficients $\phi_N$ decay sub-linearly with $N$.

It is important to study the effects of varying the length of the chain $N$ on the $\mathcal{H}_2$ norm squared in Eq. (9). We first consider the case that $w_1 = w_2 = ... = w_N = \bar{w}$, then $\xi_N = \bar{w}^{N-1}$. For this case, we see that as the length of the chain $N$ grows, $\| G \|_2^2$ in Eq. (9) will either increase or decrease, depending on whether $|\bar{w}| > c$ or $|\bar{w}| < c$, i.e., whether the magnitude of the node-to-node coupling stimulation exceeds the local suppression at each node. Intuitively, something similar happens in the case of heterogeneous $w_i$ couplings, i.e., $\| G \|_2^2$ will be large (small) when $\tilde{\xi} > c$ ($\tilde{\xi} < c$), where $\tilde{\xi} = |\xi_N|^{\frac{1}{N-1}}$ is the geometric mean of the absolute values of the node-to-node couplings along the chain with $N$ nodes.

We also derived a closed form expression for the $\mathcal{H}_2$-norm in the case in which the self-feedback coefficients at different nodes along the chain are different $c_1 \neq c_2 \neq ... \neq c_N$,

$$\| G \|_2^2 = (-1)^N \xi_N^2 \sum_{i=1}^{N} \frac{1}{2c_i} \prod_{j=1; j \neq i}^{N} \frac{1}{(c_i^2 - c_j^2)}. \tag{10}$$

See the Supplementary Material Note 1.B for our derivations. Overall, this solvable directed-chain model shows that the chain behaves as a feed-forward filter: when node-to-node stimulation exceeds local damping, signals propagate through the chain, whereas when damping dominates, the chain effectively blocks them. This pass-block behavior mirrors what occurs in more general feed-forward structures in DAG-like networks[18] and, as we demonstrate in the next section, also underpins signal transmission patterns in a large dataset of real-world networks.

## Analysis of real networks

It is natural to expect that many biological networks may have evolved in order to achieve a desired balance between passing and blocking behavior. For example, certain networks may require amplifying given input signals, while others may be used to suppress disturbances and noise. At the same time, a crucial question in technological applications is how to modify an existing network to enhance its passing versus blocking characteristics. In this context, both the directedness of the network-reflecting its asymmetry and hierarchical organization-and the functional distinction between input nodes (sources) and output nodes (sinks) are critical in determining signal propagation and amplification. Hereafter, by sources we refer to the nodes where energy, matter, or information enter the network from the environment, initiating dynamical activity that propagates through the network. In food webs, for instance, primary producers such as plants or phytoplankton act as sources by converting external energy (sunlight) into biomass[17,49,50]; in power grids, generators inject electrical energy into the network[51–57]; in connectomes, sensory or afferent regions receive external stimuli and relay signals[58–63]; in gene regulatory networks, upstream transcription factors respond to environmental cues to regulate downstream targets[17,64–67]; and in signaling pathways,

receptors at the cell membrane or intracellular sensors detect external signals and activate downstream cascades[68]. These examples underscore that identifying appropriate sources and sinks based on their biological or physical roles is essential for understanding and optimizing the passing or blocking characteristics of real-world networks.

By using the analytical tools introduced earlier, particularly the $\mathcal{H}_2$-norm, we now analyze the blocking versus passing behavior of several empirical networks. In this section, to accommodate different types of network dynamics, we consider the general nonlinear model introduced in Sec. A, $\dot{z}_i(t) = f(z_i(t)) + \sigma \sum_j \tilde{A}_{ij} g(z_i(t), z_j(t))$, $i = 1, ..., N$, where the specific forms of the functions $f$ and $g$ depend on the underlying system. These functions vary across applications-for example, they take different forms in food webs, power grids, and connectomes-as detailed in the Methods Section "Dynamical models for each network type".

In the analysis that follows, for each network model, we evaluate the Jacobian $A$ of the nonlinear system about a stable equilibrium, and use the pair $(A, B)$ to calculate the trace of the Gramian $\mathrm{Tr}(W_c)$ and the $\mathcal{H}_2$-norm. These Jacobian matrices may have entries with a negative sign. For detailed information on the nonlinear models and the evaluation of the Jacobians, see the Methods Section "Dynamical models for each network type". For each dataset, we fix the number of input nodes $m$ to its empirical (measured) value. We first choose uniformly at random 10,000 sets of $m$ nodes without repetitions, set them as input nodes, and evaluate $\mathrm{Tr}(W_c)$, the trace of the infinite-horizon continuous-time Gramian. We compare the resulting $\mathrm{Tr}(W_c)$ from randomly chosen input nodes with the $\mathrm{Tr}(W_c)$ resulting from input nodes provided by the empirical data. The comparison is performed by evaluating two measures:

$$z-\text{score} = \frac{x - \bar{x}}{s}, \quad p-\text{value} = \int_{x_{\min}}^{x_{real}} p(x)dx \bmod 0.5, \tag{11}$$

where $\bar{x}$ is the mean of the sample, $s$ is the standard deviation of the sample, $x_{\min}$ is the minimum of the sample, $x_{real}$ is the real data choice, and $p(x)$ is the probability density function such that $\int_{x_{\min}}^{x_{\max}} p(x)dx = 1$. We define

$$\text{Passing network} : \begin{cases} \text{Low } p-\text{value}, \\ \text{large positive } z-\text{score}, \end{cases}$$

$$\text{Blocking network} : \begin{cases} \text{Low } p-\text{value}, \\ \text{large negative } z-\text{score}, \end{cases}$$

and we recall that extreme (large magnitude) $z$-scores correspond to low $p$-values, indicating statistically *atypical* configurations, while small $z$-scores and high $p$-values reflect *typical* behavior. In simple terms, large positive $z$-scores correspond to networks that predominantly *pass* (amplify) inputs, large negative $z$-scores correspond to networks that predominantly *block* (attenuate) them, and values near zero indicate networks that are neither particularly passing nor particularly blocking. The results of $z$-score vs $p$-value are shown in Fig. 3 a. We consider different categories of networks: power grids for which input nodes are generators, connectomes for which input nodes are sensory neurons, molecular signaling networks for which input nodes are receptors, gene regulatory networks for which input nodes are transcription factors, pathway networks for which input nodes are extracellular ligands, and food webs for which input nodes are autotrophs (plants and algae) that sustain themselves via photosynthesis. For a detailed explanation of the real network dataset, see Supplementary Note 4. For all networks, in the absence of more detailed information, we set $C = I$. The violin plots in Fig. 3b, c show the distribution of $\mathrm{Tr}(W_c)$ along with the real network choice. These distributions and the real network choice are used to evaluate the $z$-score and $p$-value shown in panel a.

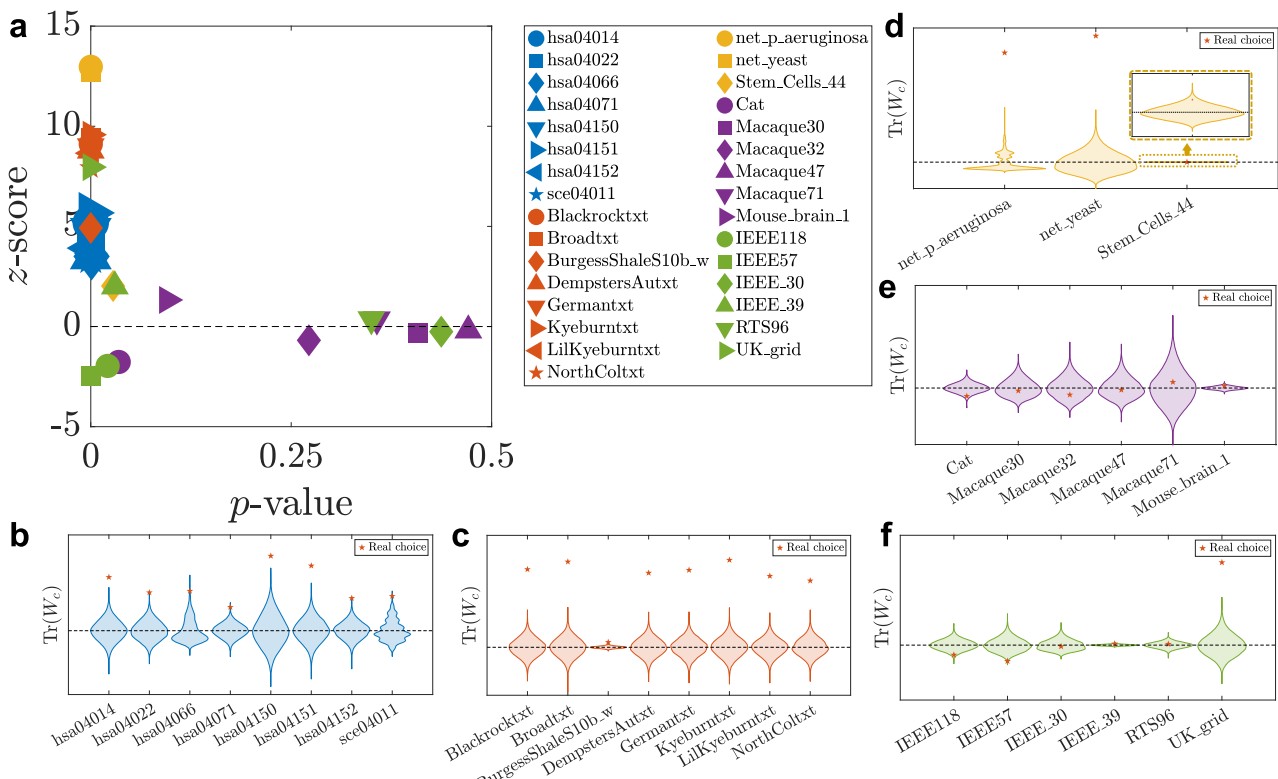

**Fig. 3 | Real data analysis.** Panel (**a**) shows the $z$-score vs $p$-value of the real data choice of the input nodes based on evaluation of the trace $\mathrm{Tr}(W_c)$ for different empirical networks (based on their Jacobian matrix $A$ and the input matrix $B$). Panels (**b**–**f**) show the distributions of $\mathrm{Tr}(W_c)$ for selected real networks over 10,000 sets of randomly chosen input nodes. The number of randomly chosen input nodes is the same as the number of input nodes from the real data. For each network, the case corresponding to the empirical selection of the input nodes is plotted as a red star. The violin plots are shifted for better visualization, such that the mean of each distribution lies on the dashed black line. Networks within the same family are plotted using the same color in different panels: (**b**) Pathway networks (blue). **c** Food Webs (red). **d** Genetic network (yellow). **e** Connectomes (purple). **f** Power Grids (green).

Figure 3 shows that in pathway networks, food webs, and genetic networks, the empirical selection of the input nodes results in passing behavior, while power grids and connectomes do not show a discernible pattern, since some networks are passing, some are blocking, and the rest are typical. The case of power grids is interesting as some of these networks are the most blocking. There are two reasons for this, the first one being that in these networks, input nodes are the generator buses where the power is injected into the grid, and the power flows on the transmission lines are given by the voltage angle differences. Typically, the generator buses either have many transmission lines to their neighbors, or a few lines with a high capacity, allowing the power flow to circulate with small angle differences and therefore ensuring the stability of the grid. One thus expects a power grid with well connected generators to be blocking in general. The second reason is that, at odds with the other types of networks considered here, power grids have a Jacobian matrix $A$ that is close to symmetric. The latter corresponds to the Jacobian of the voltage angle dynamics close to the operational state. In a grid where the resistance is negligible compared to the reactance, $A$ is symmetric. Inclusion of the transmission line losses adds a slight asymmetry to $A$, which remains close to a symmetric matrix. The violin plots confirm our conclusions while providing a more detailed view, as they display the entire distribution: atypical values appear in the tails, whereas typical values cluster near the peak, usually centered around the mean. Connectomes represent another class of networks characterized by notably strong blocking behavior. In Subsection D, we explain this in terms of their weak directionality and non-normal properties.

Complementary results are reported in Supplementary Notes 5, where the empirical adjacency matrix is used instead of the Jacobian matrix in the evaluation of the $\mathcal{H}_2$-norm, yielding comparable results. In Supplementary Note 6, we consider the largest eigenvalue of the Gramian, $\lambda_{\max}(W_c)$, as a measure of the maximum amplification rate achievable by input signals. The findings are analogous to those reported in the main text, confirming the same qualitative distinctions between passing and blocking regimes across network types. Supplementary Note 7 further presents histograms of the $\mathcal{H}_2$-norm for empirical networks, comparing two scenarios: (i) the case in which all nodes serve as input nodes, and (ii) the case in which $m$ input nodes are selected at random, with $m$ equal to the number of inputs in the empirical dataset.

### Directedness as a passing mechanism

Systems such as food webs, signaling pathways, and gene regulatory networks tend to exhibit larger $z$-scores and stronger directed hierarchical structure, being almost or strictly DAGs, consistent with their high non-normality and directional flow[22]. Non-normality, which strongly impacts the behavior of dynamical systems[19,20,24], is a prominent feature in many naturally occurring networks, as shown in Refs. 18,22. In contrast, power grids and neural networks are either blocking or typical. This is consistent with the fact that power grids tend to be symmetric, in general, and connectomes exhibit weak directedness or non-normality (see also ref. 22). Based on these structural properties, one can conjecture that most real-world networks, due to their inherent directedness, are expected to exhibit passing behavior under optimal input selection. To shed light on this conjecture, we next examine why the trace of the Gramian (or equivalently the squared $\mathcal{H}_2$-norm) varies across network classes, focusing on their directedness as captured by the Henrici index[39].

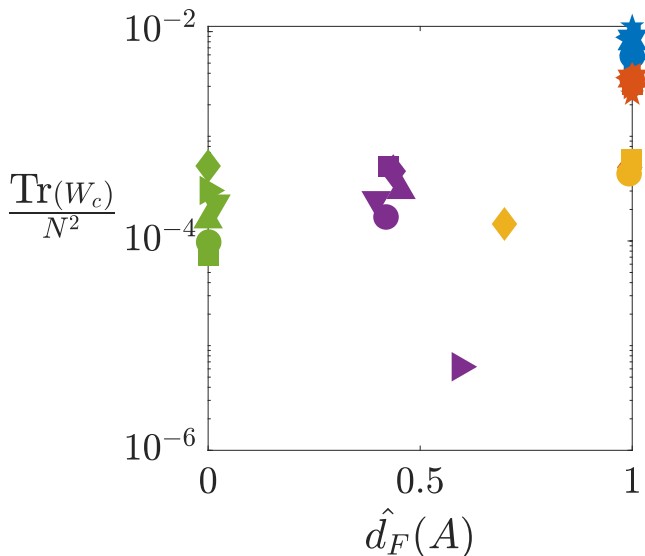

**Fig. 4 | Normalized trace of Gramian vs. the non-normality index.** The normalized Henrici index $\widehat{d_F}(A)$ of the Jacobian matrix $A$ is defined in Eq. (12). The data is displayed using the same color and marker conventions as those in Fig. 3. Pathway, Food web, Genetic, Connectome, and Power Grid networks are in blue, red, yellow, purple, and green colors, respectively.

The normalized Henrici index $\widehat{d_F}(A)$ is a measure of departure from normality[18] for a matrix $A$ based on the Frobenius norm and is defined as

$$\widehat{d_F}(A) := \frac{\sqrt{\|A\|_F^2 - \sum_{i=1}^{N}|\lambda_i|^2}}{\|A\|_F}. \quad (12)$$

Here, $\|\cdot\|_F$ denotes the Frobenius norm, and $\lambda_i$ are the (possibly) complex eigenvalues of the matrix $A$. If the matrix $A$ is normal, then $\widehat{d_F}(A) = 0$. If the matrix $A$ is triangular (highest structural non-normality), then $\widehat{d_F}(A) = 1$.

Figure 4 shows the normalized trace of the Gramian, $\mathrm{Tr}(W_c)/N^2$, versus the Henrici index $\widehat{d_F}(A)$ of the Jacobian matrix $A$. The data is generated in the same way as in Fig. 3. We observe a clear trend: higher Henrici index correlates with a higher normalized trace. Moreover, networks in the same category cluster together in this plot, revealing distinct levels of non-normality and signal amplification across categories. Notably, food webs and signaling pathways exhibit a pronounced scale separation-one to two orders of magnitude higher in normalized trace compared to other networks. This is consistent with their (almost) perfect directed acyclic graph (DAG) structure, which maximizes passing behavior along input-output paths. Interestingly, while food webs and pathways achieve this via fewer but longer paths, gene regulatory networks exhibit many very short paths, resembling a collection of directed star graphs[22]. As further detailed in the Supplementary Note 4, the normalized trace $\mathrm{Tr}(W_c)/N^2$ and the unnormalized $\mathrm{Tr}(W_c)$, both evaluated using the adjacency matrices $\tilde{A}$, in place of Jacobians, reinforce this observation by clustering gene regulatory networks closer to food webs and pathways, underscoring the role of perfect directedness in enhancing the passing property.

We will explore these structural mechanisms further in the next section.

## Directed acyclic graphs

Building on the previous findings that highlight the importance of directed acyclic graph (DAG) structures-both for their prevalence in real-world networks and their role in facilitating signal propagation-this section adopts a complementary approach to gain analytical insight into how directed paths influence the amplification of signals along input-output pathways.

We focus on open networks defined by the triplet $(A, B, C)$. The off-diagonal entries of the matrix $A$ represent the connectivity of a DAG, while the diagonal entries are negative quantities that represent the stabilizing self-feedback coefficient at each individual node. By construction, here, we take the matrix $A$ to be a Metzler matrix, i.e., a matrix for which all the off-diagonal entries are nonnegative. The quantity of interest is the $ji$-th entry of $A^{-1}$, which corresponds to the DC gain when $i$ is chosen as the input node and $j$ as the output node. Following[69] (Theorem 6.4), it is easy to calculate

$$[A^{-1}]_{ji} = \sum_{\text{directed path } \pi:i\to j}\left(\prod_{k\to l\in\pi}\frac{A_{lk}}{-A_{kk}}\right)\frac{1}{A_{jj}}. \quad (13)$$

The term $A_{lk}/(-A_{kk})$ inside the product denotes the edge gain divided by the local leak. Ultimately, the contribution of the gains and leaks along the path is divided by the local leak at the output node $j$. Since we require the matrix $A$ to have all eigenvalues with negative real parts for stability, the main diagonal of the matrix $A$ has all strictly negative entries, i.e., $A_{ii} < 0, \ \forall \ i$. Eq. (13) can be simplified to

$$[A^{-1}]_{ji} = -\sum_{\text{directed path } \pi:i\to j}\left(\prod_{k\to l\in\pi}\frac{A_{lk}}{|A_{kk}|}\right)\frac{1}{|A_{jj}|}. \quad (14)$$

It is important to highlight the intuitive interpretation of Eq. (14). We see that the passing, amplifying, or damping behavior of DAG networks as open systems depends both on the number of directed paths from input to output and on the weights of the edges that make up these paths. Each edge weight contributes to amplification if it is larger than one, or to attenuation if it is smaller than one; longer paths compound these effects. Moreover, the self-loops always contribute to damping-since in our formulation they correspond to the local stability term -and appear in the denominator of each term in the sum. Based on this reasoning, we can also understand why gene networks, which have many but very short (often one-step) paths, exhibit a considerable decrease in the normalized trace $\mathrm{Tr}(W_c)/N^2$, whereas food webs and signaling pathways, with fewer but longer weighted paths, maintain higher normalized trace values.

In conclusion, the DC gain for a directed acyclic graph, can be expressed as

$$\text{DC Gain} = 20\log_{10}\left|[A^{-1}]_{ji}\right|, \quad (15)$$

where $[A^{-1}]_{ji}$ is given explicitly in Eq. (14). Next, we present an illustrative example.

**Example 2.** Figure 5 shows a directed acyclic graph (without considering self-loops) and its associated matrix $A$.

For this network, node $i = 1$ is the input node and node $j = 5$ is the output node. There are 3 paths from node 1 to 5: $\pi_1 = (1, 2, 4, 5)$, $\pi_2 = (1, 2, 3, 4, 5)$, and $\pi_3 = (1, 3, 4, 5)$. Therefore, applying Eq. (14) results in

$$[A^{-1}]_{51} = -\frac{A_{21}A_{42}A_{54}}{A_{11}A_{22}A_{44}A_{55}} + \frac{A_{21}A_{32}A_{43}A_{54}}{A_{11}A_{22}A_{33}A_{44}A_{55}} - \frac{A_{31}A_{43}A_{54}}{A_{11}A_{33}A_{44}A_{55}}.$$

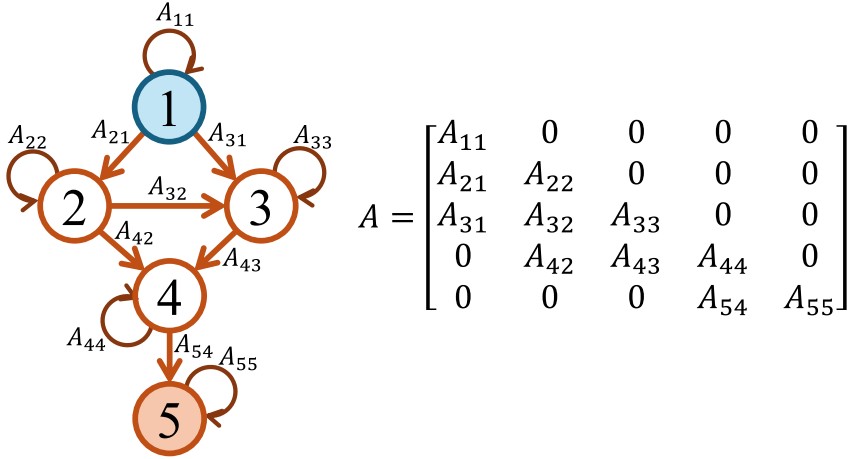

**Fig. 5 | Directed Acyclic Graph (excluding self-loops) and its corresponding Jacobian matrix $A$.** The input and the output nodes are shown in blue and orange, respectively. Self-loops are considered to provide asymptotic stability of the linear dynamics.

Note that since we assume negative self-loops for stability, then

$$[A^{-1}]_{51} = -\frac{A_{21}A_{42}A_{54}}{|A_{11}A_{22}A_{44}A_{55}|} - \frac{A_{21}A_{32}A_{43}A_{54}}{|A_{11}A_{22}A_{33}A_{44}A_{55}|} - \frac{A_{31}A_{43}A_{54}}{|A_{11}A_{33}A_{44}A_{55}|}.$$

**Frequency response of general networks**

Next, we consider general network topologies-without requiring a DAG structure-while retaining the assumption that the matrix $A$ is Metzler and has negative real part eigenvalues. We find that the slope of the magnitude plot for large frequencies ($\omega \gg$ cornering frequency) and the final phase for large frequencies follows the general relations,

$$\text{slope} = -20(d+1), \quad \text{final phase} = -90(d+1), \quad (16)$$

where $d$ is the shortest path length from the input node to the output node without considering the weights. Alternatively, $d + 1$ is the effective order of the dominant pole at the cornering frequency $c$, where $-c$ is the largest real-part eigenvalue of the matrix $A$. Figure 6 confirms our above relations for the slope, the final phase, and the cornering frequency. Note that in this example, if one considers the weights when evaluating $d$, then the path that goes from the input node to node 2 and then to the output node is no longer the shortest, while the shortest path involves visiting node 4 before node 2. The evaluation of the weighted shortest path does not match the numerical results shown in the left panels of Fig. 6, while our definition of the shortest unweighted path length matches.

## Discussion

In this work, we propose a general theory to study how networked dynamical systems behave as open systems-processing, amplifying, or attenuating environmental inputs depending on their internal structure and the specific configuration of input and output nodes. We employ the $\mathcal{H}_2$-norm as a robust and universal metric to quantify the degree of input-to-output signal amplification across a wide class of inputs, including sinusoidal, periodic, and stochastic signals. Within this framework, we show how the frequency response of an arbitrary network, including its Bode profile, is affected jointly by the underlying topology and by the choice of input and output nodes. We also examine the $\mathcal{H}_2$-norm across several empirical networks and derive new closed-form results for directed acyclic graphs and in the zero-frequency limit. In addition, we obtain an asymptotic approximation

for the trace of the Gramian in the regime dominated by strongly stable dynamics, and we uncover a modularity property of the squared $\mathcal{H}_2$-norm that applies to both input and output nodes, extending previous results that focused only on inputs.

Our theoretical results demonstrate that the signal amplification characteristics of a network are strongly influenced by structural factors such as the location of input and output nodes, the node-to-node weighted connectivity, and unweighted shortest path distances. We establish relationships between the network topology and signal transmission behavior, providing conditions under which a network tends to either pass or block signals, based on frequency response and DC gain analyses. Importantly, we identify network topologies that exhibit optimized configurations, either facilitating transmission (passing) or suppressing disturbances (blocking), reflecting functional demands of the network with its environment and constraints arising from evolution or design. The work in ref. 70 investigates open networks in discrete time; however, it does not consider their frequency response.

An empirical analysis of networks across diverse domains, including power grids, gene regulatory systems, and neural connectomes, supports our theoretical predictions and highlights the broad applicability of our framework. We find that food webs, signaling pathways, and gene regulatory circuits are often structurally organized to enhance the passing of signals, while the structure of engineered systems like power grids tends to suppress signal and noise propagation. Overall, this study lays the groundwork for systematic characterization and optimization of open networks, with potential applications in network design, control, and the interpretation of naturally occurring network dynamics. This exploration of empirical network structures, together with the dynamics they support, complements and further previous works which focused on the effect of topology and nonlinearities on the propagation of perturbations or on specific network structures that amplify or reduce fluctuations[26,27,71].

Most of the previous effort in investigating the response of networks of coupled dynamical systems to external inputs has focused on undirected networks. Many of the results obtained for undirected networks do not generalize to directed ones, which are typically more challenging to investigate. In this work, we have applied tools and obtained results that apply to generic (non-normal) network topologies and uncovered that directedness provides a fundamental passing mechanism. In particular, our data analysis reveals that directed acyclic graphs (DAGs), a topological feature that is prevalent in food webs, signaling pathways, and gene regulatory networks-exhibit strong passing behavior. This empirical observation aligns with our analytical

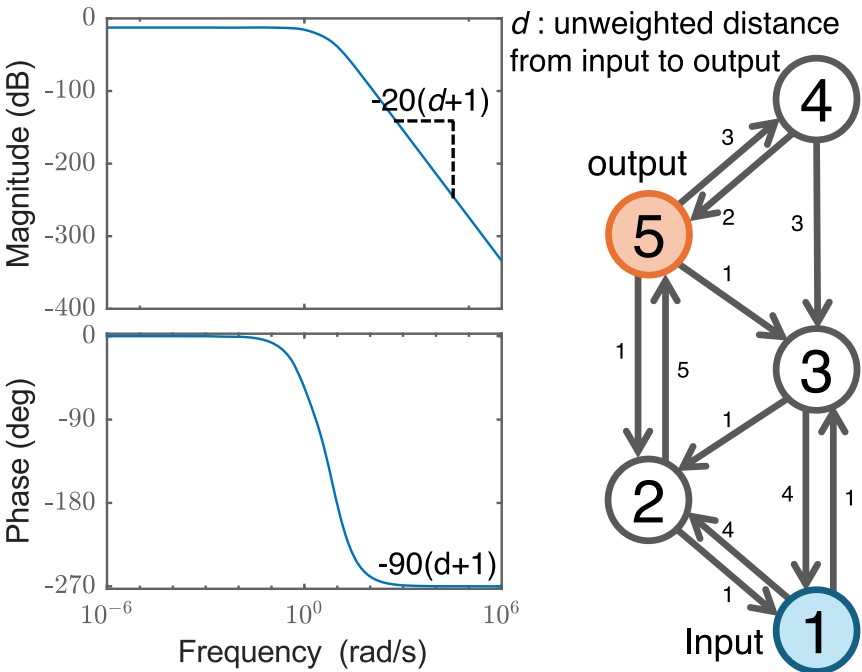

**Fig. 6 | Bode diagram for a general weighted network.** Bode plot (left panels) of the network shown on the right. The matrix $A$ defining the topology of the network is shifted $A \leftarrow A - cI$ such that the largest real part eigenvalue of the matrix $A$ is $-1$. All nodes have self-loops of weight $-c$, which are not shown for better visualization. The shortest path without considering edge weights is from the input node to node 2 and then to the output node, which has a length of $d = 2$. The slope of the magnitude after the cornering frequency $c = 1$ is slope $= -60$, and the final phase $= -270$.

results for DAGs, which demonstrate that amplification and attenuation depend explicitly on the number and length of the weighted input-output paths, as well as on attenuation effects at the node level. Taken together, these findings are consistent with the previous studies of directed, non-normal networks, where strong asymmetries in the interaction pattern generate source-sink hierarchies, suppress back-propagation, and bias flows directionality. In such architectures, which are often structurally close to acyclic and possess a dominant directed backbone with only weak feedback loops, perturbations are funneled toward terminal components and sink nodes, naturally favoring one-way signal transfer[17–19,22,25].

While our analysis captures how network structure shapes signal propagation, it relies on the linear dynamics encoded in the Jacobian of the underlying nonlinear system. As we emphasize, the LTI description reflects only the first-order behavior around a fixed point, and different nonlinear models defined on the same structure-or even different equilibria of the same model-can yield different Jacobians and thus different response patterns. This is an inherent limitation of any linearization-based approach. Nevertheless, linear analysis often provides meaningful insight into regimes where systems operate near criticality or in weakly nonlinear phases, where the Jacobian accurately captures local signal propagation. Outside these conditions, fully nonlinear effects become essential and lie beyond the scope of the present work. We also note that some of our results, in particular those in Secs. II E and II F, are limited to the case of non-negative weights associated with the network edges.

Our work provides direct insight into how to tune the $\mathcal{H}_2$-norm of an open network-either increasing it by strategically adding input or output nodes, or reducing it by removing them. This insight enables a principled approach to ranking nodes based on their contribution to the network's amplification properties, leveraging the modularity of the output Gramian. These findings have direct implications for engineering and bioengineering applications, for which achieving an optimal balance between blocking and passing behavior is often essential.

Several key questions about how complex systems respond to perturbations remain open. For instance, time-varying coupling-arising from failures or from the intrinsic evolution of the underlying network-may strongly affect signal transmission, and the impact of control actions triggered by propagating perturbations is still poorly understood. Furthermore, mounting evidence points to the pivotal role of higher-order (beyond pairwise) interactions in networked systems[72–76], raising the question of how such multi-body couplings modulate their response and resilience to perturbations. Addressing these issues represents an important direction for future work.

## Methods

### From nonlinear to linearized equations

We start from the general set of equations for the network dynamics we consider in this paper,

$$\dot{z}_i(t) = f(z_i(t)) + \sigma \sum_j \widetilde{A}_{ij}\, g(z_i(t), z_j(t)), \tag{17}$$

$i = 1, \ldots, N$, where $z_i(t) \in \mathbb{R}$ is the state of system $i$, the function $f : \mathbb{R} \to \mathbb{R}$ describes the intrinsic dynamics of each node, $g : \mathbb{R} \times \mathbb{R} \to \mathbb{R}$ the pairwise interactions, the scalar $\sigma$ the global coupling strength, and $\widetilde{A} \in \mathbb{R}^{N \times N}$ the directed adjacency matrix of the network. The specific forms of the functions $f$ and $g$ depend on the underlying network dynamics-for example, they differ for food webs, power grids, and connectomes-as detailed in the Methods Section "Dynamical models for each network type".

Linearization of Eqs. (17) around a stable fixed point $z_i^*$, $i = 1, \ldots, N$, yields the LTI dynamics,

$$\dot{x}_i(t) = A_{ii}\, x_i(t) + \sum_{j \neq i} A_{ij}\, x_j(t), \tag{18}$$

$i = 1, \ldots, N$, with $A_{ii} = f'(z_i^*) + \sigma \sum_j \widetilde{A}_{ij}\, \partial_1 g(z_i^*, z_j^*)$ capturing the local stability of the intrinsic dynamics together with the self-dependence of

the interactions, and $A_{ij} = \sigma \widetilde{A}_{ij} \, \partial_2 g(z_i^*, z_j^*)$ encoding the linearized influence of node $j$ on node $i$. Note that this equation is equivalent to our main equation (1) in the absence of inputs $u(t) = 0$.

As an example, we consider an $N = 6$-node network with adjacency matrix,

$$\widetilde{A} = \begin{bmatrix} 0 & 1 & 0 & 0 & 0 & 0 \\ 0 & 0 & 1 & 1 & 0 & 0 \\ 0 & 0 & 0 & 0 & 1 & 0 \\ 0 & 0 & 1 & 0 & 1 & 0 \\ 0 & 0 & 0 & 0 & 0 & 1 \\ 0 & 0 & 1 & 0 & 0 & 0 \end{bmatrix}.$$

With this choice of the adjacency matrix, we adopt the Michaelis-Menten model[77], a standard framework for describing the dynamics of gene-regulatory and metabolic networks,

$$\frac{dz_i(t)}{dt} = -\alpha_i z_i^a(t) + \kappa \sum_{j=1}^{N} \widetilde{A}_{ij} \frac{z_j^h(t)}{1 + z_j^h(t)}, \qquad (19)$$

$i = 1, \ldots, N$, where $z_i(t)$ represents the expression of gene $i$ at time $t$ and wthout loss of generality, we set $a = 2$, $h = 1$, $\kappa = 1$, and $\alpha_i = 1$ at all nodes. Further details on this model can be found in the Methods Section "Dynamical models for each network type".

We compute the Jacobian about the fixed point $\mathbf{z}^* = [0.692, 0.921, 0.618, 0.874, 0.618, 0.618]^T$,

$$A = \begin{bmatrix} -1.384 & 0.271 & 0 & 0 & 0 & 0 \\ 0 & -1.842 & 0.382 & 0.2847 & 0 & 0 \\ 0 & 0 & -1.236 & 0 & 0.382 & 0 \\ 0 & 0 & 0.382 & -1.748 & 0.382 & 0 \\ 0 & 0 & 0 & 0 & -1.236 & 0.382 \\ 0 & 0 & 0.382 & 0 & 0 & -1.236 \end{bmatrix}.$$

We note the similarity in the structure of the matrices $A$ and $\widetilde{A}$ above. In particular, the Jacobian $A$ is a weighted version of the network adjacency matrix $\widetilde{A}$ plus a diagonal matrix, since the off-diagonal entries of the Jacobian multiply the original entries of the adjacency matrix, and self loops might exist. When the fixed point is homogeneous across nodes, these weights reduce to a uniform rescaling of the adjacency matrix. Application of a spectral shift $A \leftarrow A - cI$, $c > 0$, only adds a uniform damping term and does not affect the eigenvectors of the matrix $A$. As a result, the network structure and qualitative dynamical response are preserved, with the shift modifying only the global timescale[7-9].

## Frequency response example

Figure 7 shows an input signal made of two sinusoidal components with different frequencies and amplitudes. The input signal enters the network through the blue node and results in the output signal measured at the red node. The output signal has the same two frequencies as the input signal, but with different amplitudes and phases. We see that the amplitude of the low-frequency input is slightly amplified (from $1 \to 1.4$), while the amplitude of the high-frequency input is damped (from $0.2 \to 0.002$).

**Technical details.** The pair $(A, B)$ is selected from our dataset for the connectome network "Macaque 30". The network has 30 nodes and has 7 input nodes. Node 4 was selected as the input (shown in blue in Fig. 7), and node 1 was selected as the output (shown in red in Fig. 7). The distance from the input to the output node is $d = 2$. The input and

the output signals are composed of the following components:

$$\text{Low Frequency input} = \sin(2\pi t), \qquad (20a)$$

$$\text{High Frequency input} = 0.2 \sin(40\pi t), \qquad (20b)$$

$$\text{Low Frequency output} = A_1 \sin(2\pi t + \phi_1), \qquad (20c)$$

$$\text{High Frequency output} = A_2 \sin(40\pi t + \phi_2), \qquad (20d)$$

where

$$A_1 = |G(2\pi ED)| = 1.41, \ \phi_1 = \angle G(2\pi ED) = -86^{\cdot}, \qquad (20e)$$

$$A_2 = |G(40\pi ED)| = 0.002, \ \phi_2 = \angle G(40\pi ED) = -251^{\cdot}, \qquad (20f)$$

where $J = \sqrt{-1}$. The amplitude and the phases of the output components can be seen in the Bode plot in the bottom panel of Fig. 7 (shown as green circles for the low-frequency signal and purple squares for the high-frequency signal). Note that magnitudes of 1.41 and 0.002 correspond to dB values $20\log_{10}(1.41) = 3$ and $20\log_{10}(0.002) = -54$, respectively.

### DC Gain
In the limit of the frequency variable approaching zero, $\omega \to 0$, the transfer function $G(J\omega)$ simplifies as follows:

$$\begin{aligned} \text{DC Gain} &= \lim_{\omega \to 0} 20 \log_{10} |G(J\omega)| \\ &= \lim_{\omega \to 0} 20 \log_{10} |C(J\omega - A)^{-1} B| \\ &= 20 \log_{10} |CA^{-1}B|, \end{aligned}$$

$$\begin{aligned} \text{Phase} &= \lim_{\omega \to 0} \angle G(J\omega) = \lim_{\omega \to 0} \angle C(J\omega - A)^{-1} B \\ &= \angle(-CA^{-1}B). \end{aligned}$$

In the case where an external signal is applied to node $i$ and the response is measured at node $j$, so that $B = \mathbf{e}_i$ and $C = \mathbf{e}_j^{\top}$, the expressions simplify to

$$\text{DC Gain} = 20\log_{10} \left| [A^{-1}]_{ji} \right| = 20\log_{10} \left| \frac{Q_{ij}}{\det(A)} \right|, \qquad (22)$$

$$\text{Phase} = \angle [A^{-1}]_{ji} = \angle \left( \frac{Q_{ij}}{\det(A)} \right), \qquad (23)$$

where $Q_{ij}$ denotes the $(j, i)$-th cofactor of $A$, defined as

$$Q_{ij} = (-1)^{i+j} \det(M_{ij}),$$

with $M_{ij}$ the submatrix of $A$ obtained by deleting row $i$ and column $j$.

### Properties of the controllability Gramian
The controllability Gramian is a symmetric and positive semi-definite matrix. The system is called controllable if and only if the matrix is nonsingular. The Gramian can be directly related to the energy required to steer the system around the state space. Assuming the initial condition for the system is at the origin, and the desired final

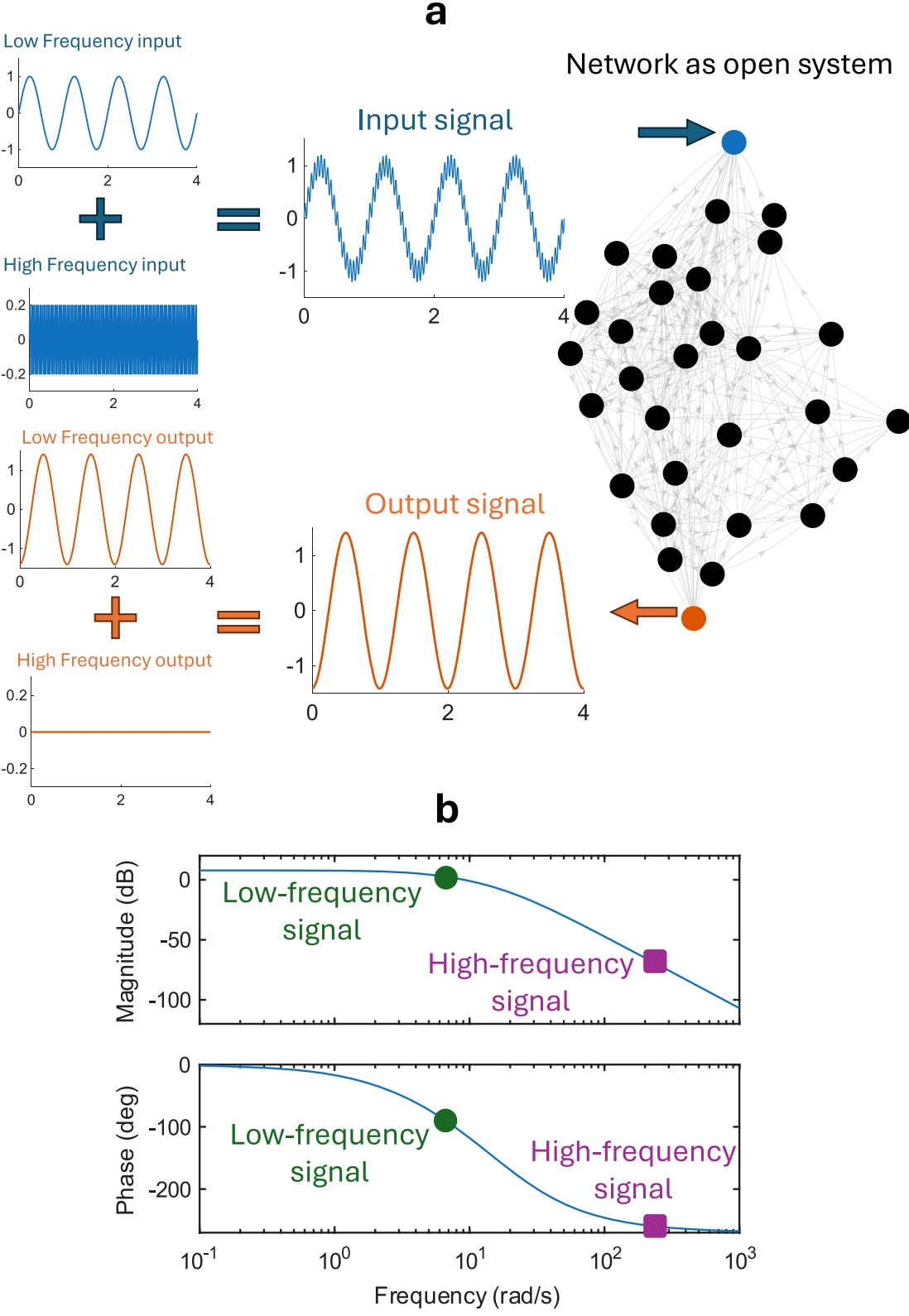

**Fig. 7 | An example based on a real connectome network.** The pair (*A*, *B*) is selected from our dataset for the connectome network "Macaque 30''. The network consists of 30 nodes, of which 7 are input nodes. Panel (**a**) shows the transmission of a low-frequency input signal and attenuation of a high-frequency signal. Panel (**b**) shows the Bode plot associated with the system.

state is $x_f$, then the solution to the minimum energy problem is

$$\text{Energy} = \frac{1}{2}\int_0^\infty \boldsymbol{u}(t)^\top \boldsymbol{u}(t) = \boldsymbol{x}_f^\top W_c^{-1} \boldsymbol{x}_f. \tag{24}$$

Here, the entries on the main diagonal of $W_c^{-1}$ represent the energy required for steering the system in canonical directions $\boldsymbol{e}_i$, i.e., $\text{Energy} = \boldsymbol{e}_i^\top W_c^{-1}\boldsymbol{e}_i = [W_c^{-1}]_{ii}$.

**Trace of continuous-time controllability Gramian.** We proceed under the assumption that all the eigenvalues of the matrix $A$ have a negative real part. We see that an important role is played by the largest real part of the eigenvalues $a = \max_i \Re(\lambda_i(A)) < 0$. If $a$ is large in magnitude, the trace of the continuous-time Gramian will be proportional to the number of input nodes $m$, i.e.,

$$\text{Tr}(W_c) = \frac{m}{2|a|}, \tag{25}$$

where $a = \max_i \Re(\lambda_i(A)) < 0$. To see this, we write $A = aI + \Delta$. From the continuous-time Lyapunov Equation we have

$$\begin{aligned}AW_c + W_cA^\top + BB^\top &= (aI + \Delta)W_c + W_c(aI + \Delta)^\top + BB^\top \\ &= 2aW_c + \Delta W_c + W_c\Delta^\top + BB^\top.\end{aligned} \tag{26}$$

Since $\|\Delta\| \ll |a|$, then $\Delta$ acts as a small perturbation. Dropping the terms containing $\Delta$, we have $W_c \approx \frac{1}{2|a|}BB^\top$ which implies

$$\text{Tr}(W_c) \approx \frac{\text{Tr}(BB^\top)}{2|a|} = \frac{m}{2|a|}. \tag{27}$$

An example of this is shown in Fig. 8, where $\text{Tr}(W_c)$ is evaluated for selected real networks. We see that when $a$, the largest real part of the eigenvalues of the matrix $A$ is large in magnitude (panel a), $\text{Tr}(W_c) \approx m/(2|a|)$, and when the magnitude is small (panel b), the trace no longer has a linear relationship with $m$.

**Modularity.** An important property of the trace of the output Gramian is modularity[78]. Modularity is analogous to linearity, which means that each element of a subset gives an independent contribution to the function value. In the particular case of the trace of the output Gramian, it is shown that the contribution of each column of the matrix $B$ to the trace is independent of the others, so one can check the contribution of each column individually and select the columns that provide desired contributions (e.g., the columns that minimize/maximize the trace). In what follows, we provide the mathematical definition of modularity.

We consider the set optimization, which is the selection of a $k$-element subset of $\mathcal{V} = \{1, 2, \ldots, M\}$ such that the function $f : 2^{\mathcal{V}} \to \mathbb{R}$ is maximized:

$$\max_{\mathcal{S} \subseteq \mathcal{V}, \, |\mathcal{S}| = k} f(\mathcal{S}). \tag{28}$$

In ref. 78, it was shown that the actuator selection for the control of linear systems (as selection of input nodes for a networked system) can be formulated as a set optimization.

**Definition 1.** (Modularity[78]). A set function $f : 2^{\mathcal{V}} \to \mathbb{R}$ is called modular if for all subsets $\mathcal{A}, \mathcal{B} \subseteq \mathcal{V}$, it holds that

$$f(\mathcal{A}) + f(\mathcal{B}) = f(\mathcal{A} \cup \mathcal{B}) + f(\mathcal{A} \cap \mathcal{B}). \tag{29}$$

If $f$ is modular, the optimization problem (28) is easily solved by simply evaluating the set function for each element, sorting the result,

and then choosing the top $k$ individual elements from the sorted list to obtain the best subset of size $k$.

The trace of the output Gramian $CW_cC^\top$ also has a modularity property with respect to the rows of the output matrix $C$. This means that $\text{Tr}(CW_cC^\top) = \sum_{j=1}^N C_{j,:}W_cC_{j,:}^\top$, where $C_{j,:} \in \mathbb{R}^{1 \times n}$ is row $j$ of the matrix $C$. To see this, we use the cyclicity and linearity of the trace and write

$$\begin{aligned}\text{Tr}(CW_cC^\top) &= \text{Tr}(W_cC^\top C) \\ &= \text{Tr}\left(W_c \sum_{j=1}^N C_{j,:}^\top C_{j,:}\right) \\ &= \sum_{j=1}^N \text{Tr}\left(W_c C_{j,:}^\top C_{j,:}\right) \\ &= \sum_{j=1}^N \text{Tr}\left(C_{j,:} W_c C_{j,:}^\top\right) \\ &= \sum_{j=1}^N C_{j,:} W_c C_{j,:}^\top\end{aligned}$$

One interesting question is how to select a set of $k$ nodes in the network as input nodes so that the trace of the output Gramian is maximized or minimized. Through the modularity property of the trace, the trace is recorded when each node is set as the input node. Then, the $k$ nodes that have resulted in the maximum/minimum trace are selected as the optimal choices for input nodes.

**Properties of the trace of the output controllability Gramian.** We define the set of input nodes $\mathcal{I}$ and the set of output nodes $\mathcal{O}$. The matrices $B$ and $C$ are uniquely defined by the set of input nodes $\mathcal{I}$ and the set of output nodes $\mathcal{O}$, as we explain next. The columns of the matrix $B$ are versors corresponding to the input nodes, i.e., each column has all zero entries except for a unique entry equal to one in the position corresponding to each one of the input nodes in $\mathcal{I}$. The rows of the matrix $C$ are versors corresponding to the output nodes, i.e., each row has a unique element equal to one in the position corresponding to each one of the output nodes in the set $\mathcal{O}$.

In accordance with the main manuscript, we define the $\mathcal{H}_2$-norm squared of the network as an open system,

$$\mathcal{H}_2^2(A, \mathcal{I}, \mathcal{O}) = \text{Tr}[CW_cC^\top], \tag{30}$$

where $W_c$ is the continuous-time controllability Gramian,

$$W_c = \int_0^{+\infty} e^{At}BB^\top e^{A^\top t}dt. \tag{31}$$

The following two properties hold:
*Additivity of the $\mathcal{H}_2$-norm with respect to the input nodes.* Consider the input node set $\mathcal{I} = \mathcal{I}_1 \cup \mathcal{I}_2$, $\mathcal{I}_1 \cap \mathcal{I}_2 = \varnothing$, then

$$\mathcal{H}_2^2(A, \mathcal{I}, \mathcal{O}) = \mathcal{H}_2^2(A, \mathcal{I}_1, \mathcal{O}) + \mathcal{H}_2^2(A, \mathcal{I}_2, \mathcal{O}). \tag{32}$$

The proof can be found in ref. 78. From (32) it follows trivially that

$$\mathcal{H}_2^2(A, \mathcal{I}, \mathcal{O}) = \sum_{i \in \mathcal{I}} \mathcal{H}_2^2(A, i, \mathcal{O}). \tag{33}$$

*Additivity of the $\mathcal{H}_2$-norm with respect to the output nodes.* Consider the output node set $\mathcal{O} = \mathcal{O}_1 \cup \mathcal{O}_2$, $\mathcal{O}_1 \cap \mathcal{O}_2 = \varnothing$, then

$$\mathcal{H}_2^2(A, \mathcal{I}, \mathcal{O}) = \mathcal{H}_2^2(A, \mathcal{I}, \mathcal{O}_1) + \mathcal{H}_2^2(A, \mathcal{I}, \mathcal{O}_2). \tag{34}$$

The proof follows from these two facts: (i) the output Gramian with output set $\mathcal{O}_i$ is equal to a minor of the Gramian obtained by

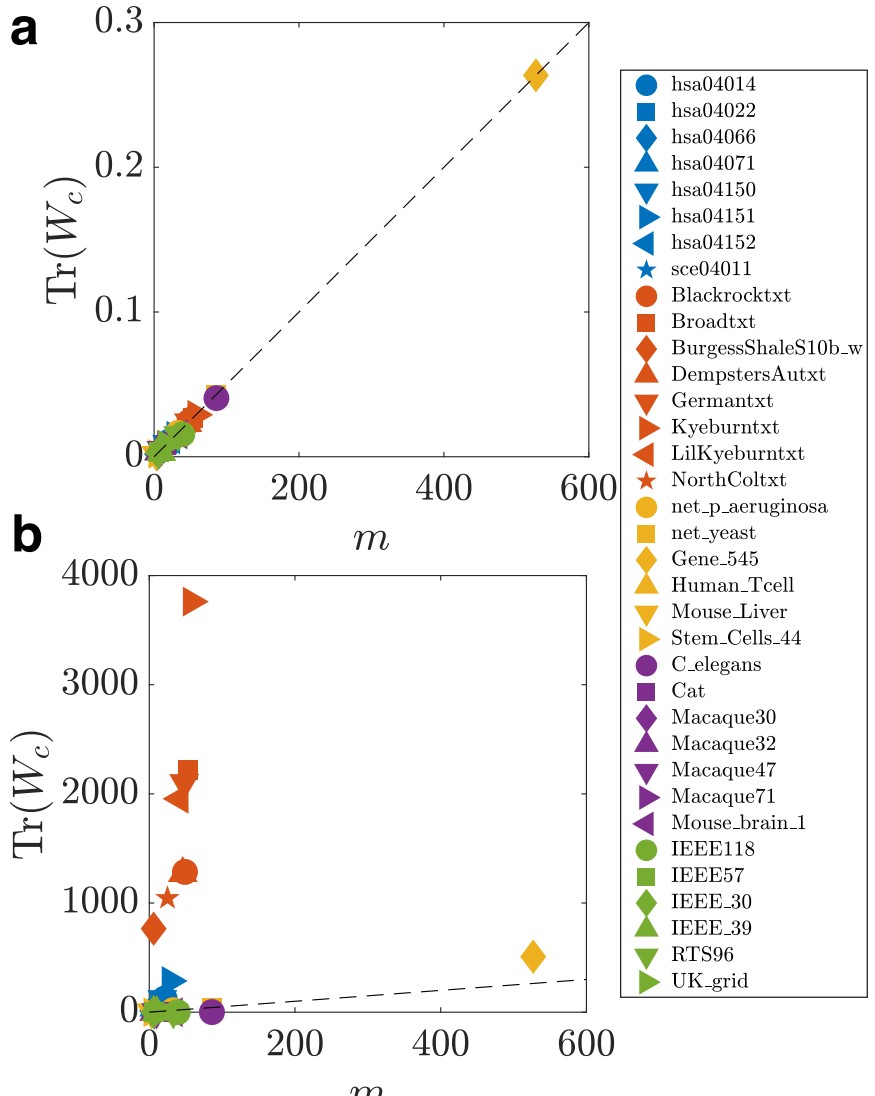

**Fig. 8 | Linear relationship between the trace and the number of input nodes.** The trace of the infinite-horizon continuous-time Gramian $\mathrm{Tr}(W_c)$ for different real networks vs the number of their input nodes $m$. In panels (**a**, **b**), the matrices $A$ are shifted so that for all networks the largest real part eigenvalues is $-10^3$ and $-1$, respectively. The dashed black line indicates the linear relationship in Eq. (25): $\mathrm{Tr}(W_c) = m/2|a| = m/2(10^3)$. The same dashed black line is maintained in Panel b for easier comparison.

selecting the rows/columns of the Gramian corresponding to the nodes in $\mathcal{O}_i$[9] and (ii) the trace of a square matrix is equal to the sum of the elements on the main diagonal. From (32) and (34) it follows trivially that

$$\mathcal{H}_2^2(A, \mathcal{I}, \mathcal{O}) = \sum_{o \in \mathcal{O}} \mathcal{H}_2^2(A, \mathcal{I}, o) = \sum_{i \in \mathcal{I}} \sum_{o \in \mathcal{O}} \mathcal{H}_2^2(A, i, o), \quad (35)$$

where $\mathcal{H}_2^2(A, i, o)$ is the individual contribution of input node $i$ and output node $o$ to the overall $\mathcal{H}_2$-norm of the network.

## Dynamical models for each network type

The matrix $A$ considered in this work is the Jacobian of the network dynamics close to a stable fixed point, for each type of network. Therefore, it depends on both the specific structure of the network and the dynamics that it supports. Here, we describe the dynamical models we used for each type of network dynamics considered in this study.

**Food webs.** We employ the generalized Lotka-Volterra model, which is a well-known model of competition and mutualism among interacting

species[18,79–81]. The governing equations of this model are:

$$\frac{dz_i(t)}{dt} = z_i(t)\left(r_i - s_i z_i(t) + \sum_{j \neq i} \tilde{A}_{ij} z_j(t)\right), \quad (36)$$

$i = 1, \ldots, N$, where $z_i(t)$ is the population of species $i$, $r_i$ are the intrinsic rates of birth if $r_i > 0$, or death if $r_i < 0$. The positive constants $s_i$ represent the finite carrying capacity of the ecosystem (limited resources) and prevent species $i$ from growing indefinitely. The interaction among species is given by the adjacency matrix $\tilde{A} = [\tilde{A}_{ij}]$, which has zeros on its main diagonal, i.e., $\tilde{A}_{ii} = 0, \forall i$.

Following[18], we assume $s_i = 1, \forall i$, for simplicity. For a proper choice of $r_i$, the point $z_i^* = 1, \forall i$, is the fixed-point of the system (if $r_i = \sum_{j \neq i} \tilde{A}_{ij}, \forall i$). Under these conditions, the Jacobian matrix $A = [A_{ij}]$ evaluated at the fixed-point $x_i^* = 1$ becomes

$$A_{ij} = \begin{cases} -\sum_{k \neq i} \tilde{A}_{ik}, & \text{if } i = j, \\ -\tilde{A}_{ij}, & \text{if } i \neq j. \end{cases} \quad (37)$$

The Jacobian $A$ has all non-positive real-part eigenvalues for networks with non-negative weights (this follows from the Gershgorin circle theorem). In practice, it may sometimes happen that the matrix $A$ has zero eigenvalues, which is undesired when calculating the Gramian (the Gramian will not converge). Therefore, we apply the spectral shift $A \leftarrow A - cI$ and properly choose $c$ so that the largest real part eigenvalue of the matrix $A$ is $-1$. The input nodes are identified, and the matrix $B$ is constructed as detailed in Supplementary Note 4A. We then perform our analysis for the pair $(A, B)$, where the matrix $B$ is constructed using knowledge of the particular selection of the input nodes from the empirical datasets.

**Connectomes.** Following[82,83], we use the Kuramoto model[36] to describe the interaction dynamics within connectome networks,

$$\frac{d\theta_i(t)}{dt} = \omega_i - \frac{K}{N}\sum_{j=1}^{N} \widetilde{A}_{ij}\sin(\theta_i(t) - \theta_j(t)), \qquad (38)$$

$i = 1, \ldots, N$, where $\theta_i \in (-\pi, \pi]$ and $\omega_i \in \mathbb{R}$ correspond, respectively, to the phase and the natural frequency at node $i$. The $N$-dimensional matrix $\widetilde{A} = [\widetilde{A}_{ij}]$ is the adjacency matrix, describing the network topology, and $N$ is the number of nodes. When $\kappa = \frac{K}{N}$ is large enough, the system typically reaches a phase-locking state where all the phases evolve at the same angular frequency given by $\Omega = N^{-1}\sum_{j=1}^{N}\omega_j$. The phase-locked state is defined as the state at which $d\theta_i(t)/dt = \Omega$, $\forall\, i$. Therefore, the phased-locked state for oscillator $i$ is $\theta_i^*(t) = \theta_i^*(0) + \Omega t$.

The Jacobian of the system $A = [A_{ij}]$ linearized about the phase-locked state is,

$$A_{ij} = \begin{cases} -\sum_k \widetilde{A}_{ik}\cos(\theta_i^*(0) - \theta_k^*(0)), & \text{if } i = j, \\ \widetilde{A}_{ij}\cos(\theta_i^*(0) - \theta_j^*(0)), & \text{if } i \neq j. \end{cases} \qquad (39)$$

In this case, the Jacobian $A$ has some of the entries in its rows equal zero, thus it has at least one eigenvalue equal to 0. Since our methodology requires asymptotic stability, we apply the spectral shift $A \leftarrow A - cI$ and properly choose $c$ so that the largest real part eigenvalue of the matrix $A$ is $-1$. The input nodes are identified, and the matrix $B$ is constructed as detailed in Supplementary Note 4B. We then perform our analysis for the pair $(A, B)$, where the matrix $B$ is constructed from knowledge of the particular selection of the input nodes derived from the empirical datasets.

In our simulations, we randomly select the natural frequencies $\omega_i$ from the standard uniform distribution (between 0 and 1). We set $K = N$, so that $\kappa = 1$.

**Power grids.** We consider a simplified swing dynamics for the voltage angles, including losses on the transmission lines. The dynamics obeys the equations[33],

$$\frac{d\theta_i(t)}{dt} = P_i - \sum_{j \neq i} B_{ij}\sin(\theta_i - \theta_j) + G_{ij}[1 - \cos(\theta_i - \theta_j)], \qquad (40)$$

$i = 1, \ldots, N$, where $\theta_i \in (-\pi, \pi]$ is the voltage phase and $P_i$ is the injected/consumed power at bus (node) $i$. The transmission line between buses $i$ and $j$ have susceptance $B_{ij}$, and conductance $G_{ij}$. The Jacobian matrix $A$ is then obtained by linearizing the dynamics around a stable operational state of the grid, which corresponds to a phase-locked state similar to Sec. IV E 2. Each fixed point depends on the topology of the grid and the dispatch of injected/consumed power $P_i$. The entries of the Jacobian are,

$$A_{ij} = \begin{cases} -\sum_{k \neq i} \widetilde{A}_{ik}\cos(\theta_i^* - \theta_k^* + \gamma_{ij}), & \text{if } i = j, \\ -\widetilde{A}_{ik}\cos(\theta_i^* - \theta_j^* + \gamma_{ij}), & \text{if } i \neq j, \end{cases} \qquad (41)$$

where $\widetilde{A}_{ij} = \sqrt{B_{ij}^2 + G_{ij}^2}$, and $\gamma_{ij} = \arctan(-G_{ij}/B_{ij})$. As the rows of the matrix $A$ in (41) sum to zero, it has at least one vanishing eigenvalue. As for the previous models, since our methodology requires asymptotic stability, we apply the spectral shift $A \leftarrow A - cI$ and properly choose $c$ so that the largest real part eigenvalue of the matrix $A$ is $-1$. The input nodes are identified, and the matrix $B$ is constructed as detailed in supplementary note 4c. We then perform our analysis for the pair $(A, B)$, where the matrix $B$ is constructed from knowledge of the particular selection of the input nodes derived from the empirical datasets.

**Genetic and pathway networks.** We use the regulatory model (Michaelis-Menten model[77]) to capture pathways and genetic dynamics. The model obeys the equations,

$$\frac{dz_i(t)}{dt} = -f_i z_i^a(t) + \kappa \sum_{j=1}^{N} \widetilde{A}_{ij}\frac{z_j^h(t)}{1 + z_j^h(t)}, \qquad (42)$$

$i = 1, \ldots, N$, where $z_i(t)$ is the gene expression at time $t$, the local dynamics $-f_i z_i^a(t)$ captures biochemical processes (where the exponent $a$ depends on the process), the matrix $\widetilde{A} = [\widetilde{A}_{ij}]$ is the weighted adjacency matrix of the network, and the term $z_j^h(t)/(1 + z_j^h(t))$ describes genetic activation, in which the exponent $h$ modulates the saturation of the term and is associated with the level of cooperation in gene regulation. For further details of the model and its nonlinear analysis, see ref. 27(Supplementary Section 4.2).

The Jacobian of the system linearized about the active fixed-point (a fixed-point other than the origin) $\mathbf{z}^* = [z_i^*]$ is $A = [A_{ij}]$ where

$$A_{ij} = \begin{cases} -f_i a z_i^{*a-1}, & \text{if } i = j, \\ \kappa \widetilde{A}_{ij}\frac{h z_j^{*h-1}}{(1 + z_j^{*h})^2}, & \text{if } i \neq j. \end{cases} \qquad (43)$$

As with other models, we apply the spectral shift $A \leftarrow A - cI$ and properly choose $c$ so that the largest real part eigenvalue of the matrix $A$ is $-1$. We then perform our analysis for the pair $(A, B)$, where the matrix $B$ is constructed from knowledge of the particular selection of the input nodes derived from the empirical datasets, see Supplementary Notes 4D and 4E.

In our simulations, we set $a = 2$, $h = 1$, $\kappa = 1$, and sample $f_i$ from the standard uniform distribution (between 0 and 1). With this choice of parameters, an active fixed-point (a fixed-point other than the origin) exists and is stable.

## Data availability
All data generated or analyzed during this study are included in this published article (and its supplementary information files).

## Code availability
The source code for the numerical simulations presented in the paper will be made available upon request, as the code is not required to support the main results reported in the manuscript.

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

## Acknowledgements

We acknowledge support from grants AFOSR FA9550-24-1-0214 and Oak Ridge National Laboratory 006321-00001A. Wai Lim Ku was supported by the National Institutes of Health grant K22HL153477

## Author contributions

A.N. worked on the theory and numerical simulations. M.A. and M.T. worked on the theory and the data analysis. W.L.K. provided assistance with biological data. F.S. worked on the theory and supervised the research. All authors contributed to writing the paper.

## Funding

## Competing interests

The authors declare no competing interests.
