## [Transparent Peer Review file · Nature Communications]

The frequency response of networks as open systems

Corresponding Author: Professor Francesco Sorrentino

Version 0:

Reviewer comments:

Reviewer #1

(Remarks to the Author)

The authors consider dynamics near equilibrium states (fixed points) such that the systems may be reasonably approximated by linear time-invariant (LTI) systems with external input vectors. The authors thus can and do take a frequency domain approach and the H_2 norm of the transfer function (matrix G) as their main analysis tool. It quantifies how a system changes the input to output (vectors). While the approach for general (also N -dimensional) LTI systems is not new, the novel focus of the analysis of the influence of the network structures and directed networks as well as the aim to distinguish passing from blocking networks constitutes a substantial advancement.

I particularly like the work also because it opens up a broad perspective on designing networks to become optimized regarding their perturbation transmission (both regarding suppressing signal propagation or optimizing passing properties).

Overall, I thus strongly recommend publication provided a number of issues are addressed.

1a) At several positions, the manuscript can be made more accessible to the prospective broad readership. I recommend, for instance, that technical terms are defined as well as intuitively explained in the manuscript, so readers not from the exact community of the authors have easier times understanding the work. I believe an improvement in this respect is particularly valuable regarding expressions that are central to the main findings, including, e.g., by explaining what DAGs, H_2 -Norm, Gramian represent intuitively and by possibly avoiding or explaining the intuition behind and meaning of other technical terms such as shape control energy, versors, z-score and p-value in the context of the messages the authors strive to convey.

1b) Possibly, an extensive use of "[controlability] Gramian" is not even necessary despite its formal role in the current work. Similarly, likely the explicit mentioning of Metzler matrices is not essential to the conveying the main message.

2) It is not exactly clear what the spectral shift operation $A \leftarrow A - cI$ means. Given a model of a real systems with given parameters, do the authors apply a model change or do they just adapt the formalism to analyze properties of the original model. I recommend to precisely state what is meant here and thus, what the matrix A in eqn. (1) represents.

3) It might be helpful to make the manuscript more easily skimmable to directly point to the main messages, not only the main technical steps. For instance, when passing to the question about how networks pass or suppress certain frequencies (before example 1) and when proceeding to real world networks (and likely some other places in the manuscript), it might be helpful to use topical questions or topical statements at the beginning of new sections and end with concluding sentences containing the core take home message of the previous section.

4) While eqn. (10) seems correct, I obtained a simpler, more easily interpretable form proportional to $d^2 / [(N-1)c^{2N-1}]$ and no further N -dependence.

5a) I recommend to not end thoughts, derivations, or sections with just reporting the mathematical results (such as Table I on page 5 and example 2 on page 8) but spend a sentence or two explaining to the broad readership what they should take home from these results.

5b) Moreover, regarding the H_2 norms reported in Table I, I do not understand the statement "as N grows, the value of the H_2 norm decreases". Would that not also depend on the value of c ? In addition, how can one interpret these results?

5c) The meaning and direct relation of non-normality to the main findings reported could be explained more clearly, both mathematically and from an intuitive point of view. Currently, while mentioned more than once, the relation to non-normality seems somewhat hidden.

6) The statistical tests performed for models of real networks are not entirely clear to me.

6a) First, please clarify exactly which aspects of the network dynamical systems are "real", i.e. from data obtained in the laboratory or otherwise, and which are assumptions about modeling. In particular, all the classes of networks tested are nonlinear systems and one needs to fix a large number of system parameters and an equilibrium point to linearize. Please state clearly which models with which types of parameters and linearized at which equilibrium and why these choices may be reasonable/why they inform us about the response dynamics of the underlying real systems.

6b) From a technical perspective, it remains unclear what the ensembles are. The main text states that sets of m nodes have been drawn repeatedly at random. What is m for each of the systems reported about in Fig. 3 and why is that reasonable? Maybe I am missing something here.

6c) Which z-scores are qualified as particularly large (positive or negative) ?

6d) Why do these choices make sense/what do they mean?

6e) Same questions c) and d) for the p values.

7) I strongly recommend to embed the core results into the literature, i.e. not only cite related work in the conclusion but actually work out parallels, novelties and remaining open questions as systematically as possible, not least relative to the state of the art reported in the introduction and mentioned above in this report.

Reviewer #2

(Remarks to the Author)

I have reviewed the manuscript titled "The Frequency Response of Networks as Open Systems". The authors tried to address a very important question in network science. They applied concepts/tools from control theory to several categories of real networks and concluded that many naturally occurring systems (e.g., food webs, signaling pathways, and gene regulatory circuits) are structurally organized to enhance the passing of signals, facilitating the efficient flow of biomass, information, or regulatory activity. By contrast, the structure of engineered systems (e.g., power grids) appears to be intentionally designed to suppress signal propagation. These findings are interesting. However, I have serious concerns about this study.

1. Non-negative weights? "All networks that we analyze are either unweighted or weighted with non-negative weights (see Supplementary Note 3)."

I think there is a fundamental problem here. The non-negative weights provided in the original network datasets do not necessarily represent the A matrix in the LTI modeling framework utilized by this study.

For TF-gene regulatory networks, we cannot assume all edges are non-negative, because biological regulation includes both activation (positive edges) and repression (negative edges), and both must be represented for accurate mechanistic or dynamical models (e.g., ODE-based gene regulatory networks). If we constrain all weights to be non-negative, we will lose the ability to represent repression, which is a core feature of gene regulation.

For food webs, in the context of carbon transfer and the Total Dependency Coefficient, all elements are non-negative, simply because they represent fractions or magnitudes of energy/carbon flow. Yet, to study food webs as dynamical systems with the LTI framework, we cannot use the carbon transfer matrix as the A matrix. Instead, we must use the Jacobian of the nonlinear population dynamics (linearized around an equilibrium). The transfer matrix is for static flow accounting, while the Jacobian governs actual dynamic interactions.

For connectomes, if we directly use the structural connectome as the A matrix, then we will have all non-negative entries. But this would only reflect anatomical wiring, not actual dynamics. If you want the A matrix to represent effective neural dynamics (how activity propagates and stabilizes), then we must allow for negative entries, because many brain regions are connected via inhibitory neurons, and even excitatory pathways can produce effective negative feedback when combined in loops. So, it is not appropriate to assume all entries in the A matrix of a connectome are non-negative when using the LTI framework for dynamics. The structural connectome adjacency is non-negative, but the dynamical Jacobian A must allow both positive and negative entries to capture excitatory and inhibitory influences.

For KEGG pathway maps, similarly, it is not appropriate to assume all elements of the A matrix for a KEGG pathway map are non-negative when studying it with the LTI framework. The A matrix is a Jacobian that must encode both positive effects (activation, production) and negative effects (inhibition, degradation).

Even for power grids, it is not appropriate to assume that all elements of the A matrix are non-negative. The Jacobian of the linearized power grid dynamics typically has negative diagonal entries (self-damping). The off-diagonal entries are usually positive (since phase differences are usually small in normal stable operating conditions), but they can become negative depending on the operating state.

That being said, directly using the edge weights listed in the various empirical network databases as the elements in the A matrix of the LTI modeling framework is not an appropriate approximation, but a fundamental mistake. All findings based on this procedure need to be revisited.

2. Where are Output nodes?

"We consider different categories of networks: power grids for which input nodes are generators, connectomes for which input nodes are sensory neurons, molecular signaling networks for which input nodes are receptors, gene regulatory networks for which input nodes are transcription factors, pathway networks for which input nodes are extracellular ligands, and food webs for which input nodes are autotrophs (plants and algae) that produce their food via photosynthesis." The choice of input nodes makes sense for all the categories of networks considered in this study. However, the choice of output nodes was not mentioned. Instead, the authors stated that "For all networks, in the absence of more detailed information, we set $C = I$." In other words, the authors selected all nodes to be output nodes. This is really a unfortunate choice, due to the data limitation. This also raises a fundamental question: Will the main conclusions on the passing and blocking behavior of various empirical networks depend on the choice of the output nodes? From Eq. (4), the H_2 -norm squared apparently depends on C . The authors should carefully address this issue.

3. Higher-order interactions?

Higher-order interactions are ubiquitous in most biological and ecological networks (food webs, connectomes, gene regulatory networks, KEGG pathways), because the underlying mechanisms are combinatorial and context-dependent. The authors should discuss if the presence of high-order interactions will affect their LTI modeling framework and their findings.

Reviewer #4

(Remarks to the Author)

The frequency response of networks as open systems - Report

The paper addresses an important and timely question: how complex networks process incoming signals. It seeks to distinguish between networks that amplify inputs - "passing networks" - and those that attenuate them - "blocking networks". Using a control engineering perspective, the authors identify the structural properties that drive each behavior and then extend their analysis to a wide range of real-world networks, classifying them accordingly. Their main result is that directed acyclic graphs (DAGs) tend to facilitate passing, whereas symmetric and loopy networks are more effective at blocking. This, they suggest, may explain why natural systems often feature DAG structures, seeking to effectively transmit information, while engineered systems frequently favor architectures that, according to the paper's analysis, suppress unwanted signal propagation.

Before turning to the detailed report, let me first offer my overall assessment of this submission.

The paper has several strengths:

1. The problem addressed is interesting and relevant.
2. The introduction, motivation and abstract are clear and suitably written for a broad readership.
3. Mathematical analysis is sound.
4. The results span a diverse set of networks from multiple domains.

The main points for improvement are:

1. While mathematically solid, the analysis feels narrower than the storyline suggests.
2. The exposition remains largely within an engineering framework. It could (and should) move beyond this to offer more "live" network dynamical insight.
3. At times the analysis is difficult to follow and not fully suited to an interdisciplinary audience.

Overall, I would be happy to see this work resubmitted after revision. Below I outline how I suggest to bring it into better shape for Nat. Comm.

Claimed generality. The study of information flow in complex networks has been rigorously advanced through a variety of mathematical approaches. Here, the authors adopt linear time-invariant (LTI) dynamics of the form given in Eq. (1). While this framework represents a central paradigm in control engineering, the paper should more clearly acknowledge its limitations in the broader study of network dynamics. Most relevant systems in this context, ranging from gene regulation to ecological interactions, are all inherently nonlinear. This is not a marginal issue: nonlinearity can fundamentally alter a network's response patterns.

The authors seek to address this limitation by noting that nonlinear systems can be locally linearized around their fixed points. While correct, this does not fully resolve the problem. Distinct nonlinear dynamics defined on the same network, or even two different fixed points of the same dynamics, can lead to very different behaviors. Yet Eq. (1) remains unchanged. In effect, the LTI formalism captures only the contribution of network structure, while overlooking the role of the system's intrinsic dynamics. However, modifying these underlying dynamics, even on a fixed structure, will change the effective weights A_{ij} and yield qualitatively different response patterns. It is therefore unsurprising that the authors' conclusions — e.g., DAGs as passing networks and symmetric ones as blocking — speak only to the effect of structure, without shedding light on the impact of nonlinear dynamics themselves.

Results and exposition. The analysis relies on two closely related characterizations of system response: the H_2 norm and

the Bode diagram. While standard in control engineering, these measures may not align with how the network science community, or the domain-specific readers, intuitively conceptualize a network's response. I therefore encourage the authors to complement their current approach with perspectives that feel more natural to network researchers. For instance: What is the average magnitude of the network's response? How is it distributed across nodes? Does it depend on the degrees of the input and output nodes? This is not to suggest that the current mathematical measures are unsuitable. Just that they are too narrow, and should be enriched and reinforced by natural network-driven measures that can highlight dimensions of the problem, which, I believe, the readers will be curious about.

On this note, reading the paper, one senses a strong disconnect from the breadth of the story, as comes up from the introduction, to the actual results, which remain abstract and - at times - may seem naïve. For example, the paper, in principle, claims that by mapping the system's response to different frequencies, one can extract its response to all general types of signals. The argument rests on the fact that complex input signals can be decomposed into their Fourier series, thus breaking them down into a sum of simpler single frequency inputs. This is clearly correct, but remains as an abstract mathematical argument. The exposition would be truly empowered if the authors go ahead and show it. For instance, simulate actual responses of real-networks, to realistic signals. Make it "come to life". Show how a complex multi-frequency input is stripped of off its high frequencies, as per the Bode diagram prediction. Demonstrate how you can use the paper's theoretical insights to generate signals that pass vs. ones that are blocked. Illustrate visually how distinct networks process similar signals - as per the paper's main prediction. As with most theoretical studies, there are two layers at play: the broad story told in the introduction and discussion, and the mathematical analysis that forms the technical core. The gap between them is bridged only through interpretation. To make that interpretation compelling, however, the figures and results must actively show the phenomena, not merely assert them. Biologists, ecologists, neuroscientists — and even physicists and engineers — may accept the mathematics, but they will only fully embrace the message once they can clearly "see" the outcomes brought to life.

Analysis. The analysis is highly "engineery" in style. This is not a problem in itself, but it does present challenges for the intended readership, many of whom are not control engineers. Even I, while not an engineer but well rooted in the network dynamics domain, found myself needing to look up certain concepts and struggling with some of the mathematical transitions. For example, Eq. (2) is not immediately intuitive, and Eq. (3) introduces the $20\log_{10}$ factor without explanation or reference to the Methods or a relevant citation (later understood to represent decibel units). The same issue arises with the definitions of the H_2 norm and the Gramian in Eqs. (4)–(6), which are presented without justification or intuitive grounding. Similarly, Eqs. (9) and (10) appear imposed rather than derived, again without explanation. To be clear, it is perfectly acceptable to relegate detailed derivations to the SI or Methods section — the reader does not need to follow every technical step. But the exposition should not force readers to decode what feel like unexplained or "magical" transitions. At minimum, the authors should provide references, intuitive explanations, or brief contextual notes that make the mathematical steps accessible and allow readers to grasp the underlying principles that enable them.

A further consequence of the paper's opaque derivations is that it is not always clear which results are canonical and which represent the paper's original contributions. Overall, the mathematics primarily involves linear algebraic manipulations grounded in well-known principles. However, the absence of references or explanatory notes makes it difficult at times to discern which steps are established and which are newly introduced in this work.

Figures. Figure 2 looks more like an illustration than an actual measurement. Axes should be labeled clearly - especially log vs. linear scale. It took me some time to understand that the slope of $-20N$ represents $1/w^N$...

Table 1 is supposed to convince us that H_2 decreases with c in a way that dominates its growth with d . I assume that this is rooted in the higher exponent applied to c . Would not it be more natural to show this as a figure, rather than a Table?

I feel that Figure 3a would be more informative if the x-axis would be presented in log-scale. Many of the symbols consolidate around small p , but as p -values go, it often matters if $p \sim 10^{-5}$ or 10^{-8} - a distinction that is fully obscured in linear scale.

Minor typos. Fig. 3 caption "the numbers of randomly input". I assume you meant "random". Text below Figure 2 " w_1, \dots, w_N ". I think it should be $w_{\{N-1\}}$.

Conclusion. As noted, the paper addresses an interesting problem and the underlying mathematics is rigorous. However, the presentation is overly narrow and formal. I strongly encourage the authors to move beyond the abstract formalism and explicitly demonstrate the broader implications and interpretations that make their results relevant to the interdisciplinary network dynamics community. It is also crucial that the paper clearly acknowledge the limitations of the LTI framework and its potentially restrictive applicability to real-world, nonlinear systems.

A final note on relevant literature. I do not want to seem like I am "fishing" for citations, but I do want to refer the authors to some relevant papers that I co-authored. I feel slightly uncomfortable doing so, however - in this case - I really believe that the authors will find them relevant, and that they can help situate the current manuscript within a broader context. I also wish to assure the authors that my assessment in round 2 of this paper, will be unaffected by their choice to cite or ignore these papers. I am truly attempting to be constructive here, not to enforce a reference :-). Over the past several years, my co-authors and I have focused on the mathematical characterization of nonlinear network dynamics - specifically, we analyzed the timescales and Jacobian structure of the system's linear response around its fixed-points. Hence, our work is intricately related to the story of the current paper. Let me mention 2 recent publications that, I think, are highly relevant and can contribute to the current paper's narrative:

1. Spatiotemporal signal propagation in complex networks. Nature Physics 15, 403–412 (2019). We consider the system's response to DC-like signals. We find that for a given (scale-free) network, the nonlinear dynamics impose distinct universality classes of propagation patterns. These universal patterns can be characterized by a single predictable exponent (θ) that predicts the intrinsic response timescales of the hub-nodes.

2. Emergent stability in complex network dynamics. Nature Physics 19, 1033–1042 (2023). We derive the Jacobian of nonlinear network dynamical systems, and find how its weights change under different nonlinear models. We show that they do not change slightly, but rather very dramatically as one transitions between nonlinear models. This touches on the relevance of the LTI framework used in the current work. For the LTI to truly approximate the nonlinear response around a fixed-point, as stated in the manuscript, one must use the appropriate Jacobian weights.

In my understanding a system's response to external signals is mediated by both the network structure and the nonlinear dynamics. The current submission primarily addresses the former, while the papers I mention above complement the analysis by elucidating how nonlinear dynamics shape response patterns. As I indicated, I do not intend to count citations in the revision round. I just think these papers can help better place the current contribution in its broader context.

I wish the authors the best of luck and look forward to reading the revised submission,
Baruch Barzel

Version 1:

Reviewer comments:

Reviewer #1

(Remarks to the Author)

Nazerian et al. have thoroughly revised their manuscript to address the multitude of comments and requests by three reviewers. In particular, they have reasonably addressed the points I have raised, better explained the technical details and complemented them with intuitive explanations and discussion where appropriate. I list below a few minor points that can still be improved.

In addition, I would like to note that I found two points raised by the two other reviewers of particular importance. First, comment 1 by reviewer 2 explicitly raised the question about the direct relation between the graph's weight matrix and the dynamical system properties resulting in the matrix elements A_{ij} , as well as the relevance of this relation. I think the authors have reasonably addressed this issue within the scope of the current work. I particularly value the new section E of the Appendix/Supplement that helps orient readers. It might still make sense to discuss in the outlook any limitations for practical applicability that might result from the mathematical conditions (non-negativity) required.

Second, reviewer comment 4.2. (along with 4.1) might be particularly valuable for connecting the general mathematical results to the broad interdisciplinary readership of Nature Communications. I believe showcasing specific examples, as attempted in the revised version of the manuscript, may make the work more accessible.
minor comments:

- The abstract is lengthy, perhaps interactions with the Editors might help further improve its reach
- non-normality (with the "-") on page 1
- I like the explicit reference on page 3 to the Methods section where A and A -tilde are explicitly related.
- I think the direct connection "bringing the results to life" (Fig. 7) should be presented and discussed within main manuscript, perhaps complemented by additional illustration/s in the main part of the work, I leave the Editors to judge this issue.
- there are a few layout/setting issues in the reference list, e.g. empty parentheses "()".

Reviewer #2

(Remarks to the Author)

The authors have successfully addressed all my previous concerns.

One minor issue: A plus sign is missing in Eq.(1) of the response letter (corresponding to Eq.(36) of the main text).

Reviewer #4

(Remarks to the Author)

I have now reviewed the revised submission and believe the authors have done an excellent job elevating the paper beyond

a narrow control-engineering perspective. I particularly appreciate the added analysis of nonlinear systems using real-world Jacobians, which provides a substantial and meaningful expansion of the paper's theoretical scope. Overall, the current version is now strengthened, and I recommend publication without further changes.

Reviewer 1

Reviewer 1 evaluation:

The authors consider dynamics near equilibrium states (fixed points) such that the systems may be reasonably approximated by linear time-invariant (LTI) systems with external input vectors. The authors thus can and do take a frequency domain approach and the H_2 norm of the transfer function (matrix G) as their main analysis tool. It quantifies how a system changes the input to output (vectors). While the approach for general (also N -dimensional) LTI systems is not new, the novel focus of the analysis of the influence of the network structures and directed networks as well as the aim to distinguish passing from blocking networks constitutes a substantial advancement.

I particularly like the work also because it opens up a broad perspective on designing networks to become optimized regarding their perturbation transmission (both regarding suppressing signal propagation or optimizing passing properties).

Overall, I thus strongly recommend publication provided a number of issues are addressed.

Answer to Reviewer 1 evaluation:

We thank the Reviewer for their thorough reading of our paper and for their constructive comments. We are indebted to all Reviewers for their relevant contributions. The paper has been revised based on the feedback received. All the changes have been highlighted in blue.

Comment Reviewer 1.1(a):

At several positions, the manuscript can be made more accessible to the prospective broad readership. I recommend, for instance, that technical terms are defined as well as intuitively explained in the manuscript, so readers not from the exact community of the authors have easier times understanding the work. I believe an improvement in this respect is particularly valuable regarding expression that are central to the main findings, including, e.g., by explaining what DAGs, H_2 -Norm, Gramian represent intuitively and by possibly avoiding or explaining the intuition behind and meaning of other technical terms such as shape control energy, versors, z -score and p -value in the context of the messages the authors strive to convey.

Answer to Reviewer 1.1(a):

We very much appreciate the feedback received by the Reviewer. We have followed the Reviewer's recommendation, and added clarifications to the text, as requested. All our changes have been highlighted in color in the revised version of the manuscript.

Comment Reviewer 1.1(b):

Possibly, an extensive use of "[controlability] Gramian" is not even necessary despite its formal role in the current work. Similarly, likely the explicit mentioning of Metzler matrices is not essential to the conveying the main message.

Answer to Reviewer 1.1(b):

We thank the reviewer for this helpful observation and we agree with the point raised. We now refer to the controllability Gramian and simply the 'Gramian' throughout the text. We have removed our reference to Metzler matrices and only kept it in one place, where we felt it was necessary to convey our message. We believe this change improves the clarity and focus of the presentation.

Comment Reviewer 1.2:

It is not exactly clear what the spectral shift operation $A \leftarrow A - cI$ means. Given a model of a real systems with given parameters, do the authors apply a model change or do they just adapt the formalism to analyze properties of the original model. I recommend to precisely state what is meant here and thus, what the matrix A in eqn. (1) represents.

Answer to Reviewer 1.2:

The Reviewer has an excellent point here. We have followed an approach that is common throughout the literature, see e.g. G. Yan, G. Tsekenis, B. Barzel, J.-J. Slotine, Y.-Y. Liu, and A.-L. Barabasi, *Nature Physics* 11, 779 (2015); J. Gao, Y.-Y. Liu, R. M. D'souza, and A.-L. Barabasi, *Nature communications* 5, 1 (2014); I. Klickstein, A. Shirin, and F. Sorrentino, *Nature communications* 8, 15145 (2017). We start from the original system equation $\dot{x}_i = -A_{ii}x_i + \sum_j A_{ij}x_j$, where the scalar A_{ii} reflects the local stability of the intrinsic dynamics and the off diagonal entries A_{ij} encode the network interactions. Many datasets include information on the network interactions, i.e., the off diagonal entries of the matrix A , but not on the individual rate of stability of the dynamics at each one of the network nodes given by c . In fact, this information is typically unavailable. The assumption often used in the literature, e.g. the papers mentioned above, is that, although c is unknown, it needs to be large enough to make the matrix $A - cI$ Hurwitz (the dynamics stable.) We take this view in this paper.

We emphasize the main motivation for the shift $A \leftarrow A - cI$ is to make sure that the dynamics is stable. Since information on the individual dynamics of each one of the uncoupled systems is often unavailable, the spectral shift operation is introduced to guarantee stability of the matrix A . To make the results of the analysis comparable from network to network (from dataset to dataset) we have chosen c for different networks such that the largest eigenvalue of $A - cI$ is equal to -1 . This is also done similarly to the above mentioned papers from the literature. In the revised version of the paper we have better clarified this aspect.

Comment Reviewer 1.3:

It might be helpful to make the manuscript more easily skimmable to directly point to the main messages, not only the main technical steps. For instance, when passing to the question about how networks pass or suppress certain frequencies (before example 1) and when proceeding to real world networks (and likely some other places in the manuscript), it might helpful to use topical questions or topical statements at the beginning of new sections and end with concluding sentences containing the core take home message of the previous section.

Answer to Reviewer 1.3:

We thank the Reviewer for this valuable suggestion and acknowledge it as a limitation of the original version of the paper. In the revised manuscript, we have strengthened the narrative flow by briefly recapping the main idea at the end of each section and clearly explaining what the next section will do and why it follows. In this way, the sections are better connected, the transitions feel more natural, and the overall story of the paper is easier to follow for a broad audience. These changes also help readers identify the key ideas and take-home messages more readily.

Comment Reviewer 1.4:

While eqn. (10) seems correct, I obtained a simpler, more easily interpretable form proportional to $d^2/[(N-1)c^{(2N-1)}]$ and no further N-dependence.

Answer to Reviewer 1.4:

We thank the Reviewer for proposing an alternative formula. We have tried to simplify our formulation, but we could not find a simpler form than the one that is included in the manuscript. We have now added a proof for our formula in the Supplementary Information, Note 1A. Moreover, we have derived an additional formula for the case in which the self-coupling loops at different nodes along the chain are different, see the Supplementary Information, Note 1B.

Comment Reviewer 1.5(a):

I recommend to not end thoughts, derivations, or sections with just reporting the mathematical results (such as Table I on page 5 and example 2 on page 8) but spend a sentence or two explaining to the broad readership what the they should take home from these results.

Answer to Reviewer 1.5:

We thank the Reviewer for this useful recommendation. With the same intention of making the manuscript more accessible to a broad readership, we have expanded the commentary accompanying our mathematical derivations and examples. In particular, we now add brief explanations of the main take home messages after key results, as is the case for our discussion of the z -score and p -value analysis, where we explicitly explain what large positive, large negative, and near zero values mean for passing or blocking behavior, so that readers can more easily understand the significance of the calculations.

In addition, and in line with the comments of the third Reviewer, we have simplified several aspects of the presentation by removing terminology that was not essential for our purposes, streamlining the definitions of the matrices B and C , and smoothing parts of the formalism to improve clarity. These changes, together with the added explanatory comments, make the mathematical development and the examples easier to follow for a general audience.

Comment Reviewer 1.5(b):

Moreover, regarding the H_2 norms reported in Table I, I do not understand the statement "as N grows, the value of the H_2 norm decreases". Would that not also depend on the value of c ? In addition, how can one interpret these results?

Answer to Reviewer 1.5(b):

We are highly indebted to the Reviewer for pointing this out. The Reviewer's excellent comment prompted us to reconsider our conclusions. For each value of N , the \mathcal{H}_2 -norm squared depends on both the amplification rates w_i 's and the shift variable c . As we have mentioned before, the shift variable c is selected so as to ensure that the system is asymptotically stable with a given rate of asymptotic stability. This is done independent of the network size. In the revised paper (page 5) we added this: 'It is important to study the effects of varying the length of the chain N on the \mathcal{H}_2 norm squared. We first consider the case that $w_1 = w_2 = \dots = w_N = \bar{w}$, then $\xi_N = \bar{w}^{N-1}$. For this case, we see that as the length of the chain N grows, $\|G\|_2^2$ will either increase or decrease, depending on whether $|\bar{w}| > c$ or $|\bar{w}| < c$, i.e., whether the magnitude of the node-to-node coupling stimulation exceeds the local suppression at each node.' To be precise, one also needs to consider the decay that can be seen from the coefficients: $1/2, 1/4, 3/16, 5/32, 35/265, \dots$. These are now shown in Table I for different values of the system order N .

The discussion that accompanies Table 1 has been modified in the revised manuscript—all changes highlighted in color. The purpose of the table is to show that the coefficients that multiply the powers of c and \bar{w} decay 'slowly' with the number of nodes N , i.e., in a sublinear way. Because of the slow decay of the coefficients, whether the \mathcal{H}_2 norm increases or decreases with N is determined by the balance between the self suppression coefficient c and the node-to-node average stimulation coefficient $|\bar{w}|$. This has been now clarified and commented on in the revised manuscript. Following the Reviewer's feedback, we have now removed the sentence 'as N grows, the value of the H_2 norm decreases' and added an explanation for how one could interpret our results. Moreover, based on the Reviewer's feedback we have also extended our analysis to consider the cases of different self loops c_i 's (in addition to different couplings w_i 's)—see Supplementary Note 1B.

Comment Reviewer 1.5(c):

The meaning and direct relation of non-normality to the main findings reported could be explained more clearly, both mathematically and from an intuitive point of view. Currently, while mentioned more than once, the relation to non-normality seems somewhat hidden.

Answer to Reviewer 1.5(c):

We thank the Reviewer for this insightful comment. We have now revised the Introduction to state the role of non-normality explicitly, both mathematically and intuitively. In particular, we added a clear description of what constitutes a non-normal network and why this property is central to our results. As shown in the data analysis of Ref. [1], many—and in several contexts most—real-world networks display a highly directed, hierarchical architecture, with an overwhelming presence of source and sink nodes and very few cycles. From a more mathematical perspective, this is captured by a measure of the “distance” from the largest underlying DAG subgraph, which quantitatively assesses how close a given network is to being acyclic and, in turn, reveals its strong similarity to Directed Acyclic Graphs. We also briefly connect this picture to existing information-theoretic characterizations: in Ref. [2], the entropy rate of a random walk decreases as the network becomes more directed or non-normal, reflecting more localized, one-way flow toward terminal components; in Ref. [3], the entropy production increases with directedness, quantifying the irreversibility of such flows. Both viewpoints are consistent with, and complementary to, our current framework, where strong directedness and non-normality underpin the enhanced passing properties and one-way signal transmission captured by a large \mathcal{H}_2 norm. We now highlight in the Introduction that this topological and dynamical structure has been a key motivation for our study, as it naturally leads to the transient amplification and directional propagation mechanisms underpinning our main findings.

Comment Reviewer 1.6:

The statistical tests performed for models of real networks are not entirely clear to me.

Answer to Reviewer 1.6:

We thank the Reviewer for this comment. We have revised the "Analysis of Real Networks" section to explicitly detail our methodology.

In summary, our statistical test compares the amplification (trace of the Gramian) of the real input node configuration against a null distribution generated by selecting the same number of input nodes uniformly at random. This comparison allows us to determine if the specific biological or technological configuration passes or blocks signals significantly more than a random configuration would.

We have provided detailed clarifications regarding the models used, the ensemble generation, and the statistical metrics (z-scores and p-values) in the specific point-by-point responses to comments 1.6(a) through 1.6(e) below.

Comment Reviewer 1.6(a):

First, please clarify exactly which aspects of the network dynamical systems are "real", i.e. from data obtained in the laboratory or otherwise, and which are assumptions about modeling. In particular, all the classes of networks tested are nonlinear systems and one needs to fix a large number of system parameters and an equilibrium point to linearize. Please state clearly which models with which types of parameters and linearized at which equilibrium and why these choices may be reasonable/why they inform us about the response dynamics of the underlying real systems.

Answer to Reviewer 1.6(a):

We thank the reviewer for their comment. Based on the feedback received we have now added a new section to the Methods, ‘E. Dynamical models for each network type’ which addresses individually all the nonlinear network models we have considered, their linearization, and the exact procedure we have followed to compute the \mathcal{H}_2 norm, etc.

For the power grids, we considered a simplified swing dynamics where only the first order voltage phase dynamics is taking into account. The line losses is accounted for in some of the power grid networks, which make the Jacobian asymmetrical. The fixed point correspond to the operational state of the grid where the voltage phases have reached a steady state such that the power flows on the lines are constant over time. The Jacobian correspond to the local dynamics of the grid around this state. We added a section in the Methods where the dynamics for each type of network is given.

Comment Reviewer 1.6(b):

Form a technical perspective, it remains unclear what the ensembles are. The main text states that sets of m nodes have been drawn repeatedly at random. What is m for each of the systems reported about in Fig. 3 and why is that reasonable? Maybe I am missing something here.

Answer to Reviewer 1.6(b):

Each real dataset that we have in the paper comes with a certain number of nodes that interact with the environment. We denote this number of nodes by m in the main text. For example, the real connectome network "C-elegans" has 283 nodes, 86 of which interact with the environment, so here $m = 86$. Another network, such that the foodweb "Broadtxt" has a total of 94 nodes, 53 of which interact with the environment, so here $m = 53$. In the Supplemental Information Table I 'Real networks information', we provide the complete information on all the networks, including the number of inputs m (the fifth column from the left).

When performing the statistical analysis in the main text (Fig. 3), we assess how the real choice of these m nodes affects the passing/blocking nature of the network. Therefore, we randomly select many sets of m nodes, set them as the input nodes, calculate the trace of the controllability Gramian, and evaluate the p -value and the z -score to determine if the original real input nodes, in fact, result in a passing or a blocking behavior.

The reason we do not change the number of input nodes when applying the statistical analysis is that the value of the trace of the controllability Gramian depends on m (often a heavy dependence, as denoted in Fig. 8 of the main manuscript). Therefore, for a fair comparison, we fix the number of input nodes m for each dataset to the number of input nodes provided by the empirical data source.

Comment Reviewer 1.6(c):

Which z -scores are qualified as particularly large (positive or negative) ?

Answer to Reviewer 1.6(c):

We thank the reviewer for their comment. A "particularly large" z -score is one that is statistically significant. In our manuscript, a z -score with a magnitude greater than 1.96 is considered significant at the $p < 0.05$ level.

Comment Reviewer 1.6(d):

Why do these choices make sense/what do they mean?

Answer to Reviewer 1.6(d):

We thank the reviewer for their comment. The meaning of a large z -score is related to the classification of networks as 'passing' or 'blocking'. A large positive z -score indicates that the real choice of input nodes results in a value of the trace $Tr(W_c)$ value that is significantly higher than random. We define this as a 'passing network', implying that network is structured to enhance input signals. A large negative z -score indicates that the real choice of input nodes results in a value of the trace $Tr(W_c)$ that is significantly lower than random. We define this as a 'blocking network', implying that the network is structured to suppress input signals.

Comment Reviewer 1.6(e):

Same questions c) and d) for the p values.

Answer to Reviewer 1.6(e):

We thank the reviewer for this comment. (c) Which p-values are qualified as significant? We qualified a p -value as significant if it is less than 0.05 ($p < 0.05$). This threshold is the probabilistic counterpart to the z-score magnitude of 1.96 mentioned in our answer to 1.6(c). (d) Why do these choices make sense/what do they mean? This choice ($p < 0.05$) represents the alpha level (α), which is the pre-defined probability threshold for controlling the Type I error rate (i.e., a "false positive"). In our study, the null hypothesis (H_0) is that the $Tr(W_c)$ value from the real input nodes is generated by random choices of the nodes. When we observe a $p < 0.05$, it means there is less than a 5% probability that our observed $Tr(W_c)$ value could have occurred by a random choice alone. This allows us to reject the null hypothesis and conclude that our finding is significant. We classify network as either 'passing' or 'blocking' based on this statistical analysis.

Comment Reviewer 1.7:

I strongly recommend to embed the core results into the literature, i.e. not only cite related work in the conclusion but actually work out parallels, novelties and remaining open questions as systematically as possible, not least relative to the state of the art reported in the introduction and mentioned above in this report.

Answer to Reviewer 1.7:

We thank the reviewer for this helpful suggestion. In the revised manuscript, we have made a more deliberate effort to embed our core results within the existing literature. In both the Introduction and the Conclusions, we now draw clearer parallels with previous work on directed and non-normal networks, emphasizing how our frequency-response and \mathcal{H}_2 -based framework complements established results on source-sink hierarchies, looplessness, and biased downstream flows, and how our analysis of directed acyclic graphs fits naturally into this broader structural picture. In addition, following the recommendation to articulate remaining open questions more systematically, we have expanded the final part of the Conclusions. There we now discuss how time-varying coupling—for example, due to failures or slow structural evolution—and control actions triggered by propagating perturbations may fundamentally reshape signal transmission, and how higher-order (beyond pairwise) interactions, as highlighted in recent work on higher-order networks, are expected to modulate both response and resilience. These aspects are now explicitly framed as natural extensions of our framework and as concrete directions for future work.

Reviewer 2

Reviewer 2 evaluation:

I have reviewed the manuscript titled “The Frequency Response of Networks as Open Systems”. The authors tried to address a very important question in network science. They applied concepts/tools from control theory to several categories of real networks and concluded that many naturally occurring systems (e.g., food webs, signaling pathways, and gene regulatory circuits) are structurally organized to enhance the passing of signals, facilitating the efficient flow of biomass, information, or regulatory activity. By contrast, the structure of engineered systems (e.g., power grids) appears to be intentionally designed to suppress signal propagation. These findings are interesting. However, I have serious concerns about this study.

Answer to Reviewer 2 evaluation:

We thank the Reviewer for their thorough reading of our paper and for their constructive comments. We are indebted to all Reviewers for their relevant contributions. The paper has been revised based on the feedback received. All the changes have been highlighted in blue.

Comment Reviewer 2.1:

Non-negative weights? “All networks that we analyze are either unweighted or weighted with non-negative weights (see Supplementary Note 3).” I think there is a fundamental problem here. The non-negative weights provided in the original network datasets do not necessarily represent the A matrix in the LTI modeling framework utilized by this study.

For TF-gene regulatory networks, we cannot assume all edges are non-negative, because biological regulation includes both activation (positive edges) and repression (negative edges), and both must be represented for accurate mechanistic or dynamical models (e.g., ODE-based gene regulatory networks). If we constrain all weights to be non-negative, we will lose the ability to represent repression, which is a core feature of gene regulation.

For food webs, in the context of carbon transfer and the Total Dependency Coefficient, all elements are non-negative, simply because they represent fractions or magnitudes of energy/carbon flow. Yet, to study food webs as dynamical systems with the LTI framework, we cannot use the carbon transfer matrix as the A matrix. Instead, we must use the Jacobian of the nonlinear population dynamics (linearized around an equilibrium). The transfer matrix is for static flow accounting, while the Jacobian governs actual dynamic interactions.

For connectomes, if we directly use the structural connectome as the A matrix, then we will have all non-negative entries. But this would only reflect anatomical wiring, not actual dynamics. If you want the A matrix to represent effective neural dynamics (how activity propagates and stabilizes), then we must allow for negative entries, because many brain regions are connected via inhibitory neurons, and even excitatory pathways can produce effective negative feedback when combined in loops. So, it is not appropriate to assume all entries in the A matrix of a connectome are non-negative when using the LTI framework for dynamics. The structural connectome adjacency is non-negative, but the dynamical Jacobian A must allow both positive and negative entries to capture excitatory and inhibitory influences.

For KEGG pathway maps, similarly, it is not appropriate to assume all elements of the A matrix for a KEGG pathway map are non-negative when studying it with the LTI framework. The A matrix is a Jacobian that must encode both positive effects (activation, production) and negative effects (inhibition, degradation). Even for power grids, it is not appropriate to assume that all elements of the A matrix are non-negative. The Jacobian of the linearized power grid dynamics typically has negative diagonal entries (self-damping). The off-diagonal entries are usually positive (since phase differences are usually small in normal stable operating conditions),

but they can become negative depending on the operating state. That being said, directly using the edge weights listed in the various empirical network databases as the elements in the A matrix of the LTI modeling framework is not an appropriate approximation, but a fundamental mistake. All findings based on this procedure need to be revisited.

Answer to Reviewer 2.1:

We thank the reviewer for this important feedback. In the revised manuscript, we now work explicitly with the Jacobian matrix obtained by linearizing the underlying (nonlinear) network dynamics about a fixed point from the very beginning, rather than using the raw adjacency matrix. The Reviewer will find extensive changes about this throughout the main manuscript, the Methods, and Supplementary Information. We also point out that, topologically, the Jacobian inherits the same sparsity pattern as the structural adjacency: the adjacency specifies which pairs of nodes can interact, while the Jacobian assigns signed weights (positive or negative) to those links, reflecting activation, inhibition, excitation, damping, etc. In this sense, the Jacobian “mimics” the adjacency at the level of which edges are present, but it refines it by providing the effective linear coupling strengths, including negative entries where appropriate. This is also demonstrated with an example in the new Methods Section A.

For each family of empirical networks, we use different nonlinear models for the network dynamics: a standard nonlinear model—generalized Lotka–Volterra dynamics for food webs, a Kuramoto phase-oscillator model for connectomes, and a Michaelis–Menten–type regulatory model for pathway and genetic networks—now described in detail in the revised manuscript, see in particular the Methods Section E. With this formulation, our analysis no longer assumes non-negative dynamical weights, i.e., the entries of the Jacobian matrix A are both positive and negative, and all our new results are obtained using the Jacobian as the matrix A in the LTI framework. Accordingly, we have updated Fig. 3 and Fig. 4 of the main manuscript to show the results obtained with the Jacobian matrices of the real networks. At the same time, we have moved the ‘old’ Fig. 3 and Fig. 4 to the Supplementary Information for a comparison. The results appear to be similar. The new Fig. 3 of the main manuscript is also included here for easier access, see Fig. 1 below.

With regards to the specific comments in reference to food webs, power grids, connectomes, and genetic networks, we now discuss each one of these cases in detail in the new Methods Section E. Below we clarify the main steps of our approach and results for different types of networks.

Food Webs:

We see that Food Webs have large z -scores and low p -values, making them passing networks. The details of evaluating the Jacobian are as follows.

We considered the generalized Lotka–Volterra model, which is a well-known model of competition and mutualism among interacting species [1, 4–6]. The governing equations of this model are:

$$\frac{dx_i(t)}{dt} = x_i(t) \left(r_i - s_i x_i(t) \sum_{j \neq i} \tilde{A}_{ij} x_j(t) \right), \quad i = 1, \dots, N, \quad (1)$$

where $x_i(t)$ is the population of species i , r_i are the intrinsic rates of birth if $r_i > 0$, or death if $r_i < 0$. The positive constants s_i represent the finite carrying capacity of the ecosystem (limited resources) and prevent species i from growing indefinitely. The interaction among species is given by the adjacency matrix $\tilde{A} = [\tilde{A}_{ij}]$, where it is assumed to have zeros on its main diagonal, i.e., $\tilde{A}_{ii} = 0, \forall i$.

Following [1], we assume $s_i = 1, \forall i$, for simplicity. For a proper choice of r_i , the point $x_i^* = 1, \forall i$, is the fixed-point of the system (if $r_i = \sum_{j \neq i} \tilde{A}_{ij}, \forall i$). Under these conditions, the Jacobian matrix $A = [A_{ij}]$ evaluated at the fixed-point $x_i^* = 1$ becomes

$$A_{ij} = \begin{cases} -\sum_{k \neq i} \tilde{A}_{ik}, & \text{if } i = j, \\ -\tilde{A}_{ij}, & \text{if } i \neq j. \end{cases} \quad (2)$$

Connectomes

We calculate the Jacobian of a nonlinear system and use the Jacobian for our statistical analysis of Connectomes. We use the Kuramoto model as follows:

$$\frac{d\theta_i(t)}{dt} = \omega_i + \frac{\sigma}{N} \sum_{j=1}^N \tilde{A}_{ij} \sin(\theta_j(t) - \theta_i(t)), \quad i = 1, \dots, N, \quad (3)$$

FIG. 1: **Real data analysis.** Panel a shows the z -score vs p -value of the real data choice of the input nodes based on evaluation of the trace (W_c) of real network data (Based on their Jacobian matrices J and the input matrix B). Panels b-f show the distributions of (W_c) for selected real networks over 10,000 sets of randomly chosen input nodes.

The number of randomly input nodes is the same as the number of input nodes from the real data. For each network, the choice of real input nodes is plotted as a red star. The violin plots are shifted for better visualization, such that the mean of each distribution lies on the dashed black line. Networks within the same family are plotted using the same color in different panels: b: Pathway networks (blue), c: Food Webs (orange), d: Genetic network (yellow), e: Connectomes (purple), and f: Power Grids (green).

where $\theta_i(t)$ is the angle of the phase oscillator of node i , and ω_i is the natural frequency of node i . The N -dimensional matrix $\tilde{A} = [\tilde{A}_{ij}]$ is the adjacency matrix, describing the network topology, and N is the number of nodes. The coupling strength is denoted by $\sigma > 0$. In this example, we set $\sigma = N$.

The phase-locked state is defined as when $d\theta_i(t)/dt = \Omega, \forall i$. Therefore, the phased-locked state equation becomes:

$$\Omega = \omega_i + \sum_{j=1}^N \tilde{A}_{ij} \sin(\theta_j^*(0) - \theta_i^*(0)), \quad i = 1, \dots, N, \quad (4)$$

where $\theta_i^*(t) = \theta_i^*(0) + \Omega t$ is the phased-locked state of node i .

After the phase oscillators reach the phase-locked state, the Jacobian $A = [A_{ij}]$ of the system linearized about the phase-locked state is,

$$A_{ij} = \begin{cases} -\sum_k \tilde{A}_{ik} \cos(\theta_j^*(0) - \theta_i^*(0)), & \text{if } i = j, \\ \tilde{A}_{ij} \cos(\theta_j^*(0) - \theta_i^*(0)), & \text{if } i \neq j. \end{cases} \quad (5)$$

For the choice of natural frequencies, we randomly select ω_i from a uniform distribution between 0 and 1.

Pathways and genetic networks

We use the regulatory model (Michaelis-Menten model [7]) to capture pathways and genetic dynamics. The model is defined as

$$\frac{dx_i(t)}{dt} = -f_i x_i^a(t) + \kappa \sum_{j=1}^N \tilde{A}_{ij} \frac{x_j^h(t)}{1 + x_j^h(t)}, \quad i = 1, \dots, N, \quad (6)$$

where $x_i(t)$ is the gene expression at time t , the local dynamics $-f_i x_i^a(t)$ captures biochemical processes, the matrix $\tilde{A} = [\tilde{A}_{ij}]$ is the weighted adjacency matrix, and $x_j^h(t)/(1 + x_j^h(t))$ describe genetic activation. For further details of the model and its nonlinear analysis, see [8, Supplementary Section 4.2].

In our simulations, we set $a = 2$, $h = 1$, $\kappa = 1$, and sample f_i from the standard uniform distribution (between 0 and 1). With this choice of parameters, an active fixed-point (a fixed-point other than the origin) exists and is stable.

The Jacobian $A = [A_{ij}]$ of the system linearized about the active fixed-point is,

$$J_{ij} = \begin{cases} -f_i a y_i^{a-1}, & \text{if } i = j, \\ \kappa A_{ij} \frac{h y_j^{h-1}}{(1 + y_j^h)^2}, & \text{if } i \neq j. \end{cases} \quad (7)$$

Comment Reviewer 2.2:

Where are Output nodes? “We consider different categories of networks: power grids for which input nodes are generators, connectomes for which input nodes are sensory neurons, molecular signaling networks for which input nodes are receptors, gene regulatory networks for which input nodes are transcription factors, pathway networks for which input nodes are extracellular ligands, and food webs for which input nodes are autotrophs (plants and algae) that produce their food via photosynthesis.” The choice of input nodes makes sense for all the categories of networks considered in this study. However, the choice of output nodes was not mentioned. Instead, the authors stated that “For all networks, in the absence of more detailed information, we set $C = I$.” In other words, the authors selected all nodes to be output nodes. This is really a unfortunate choice, due to the data limitation. This also raises a fundamental question: Will the main conclusions on the passing and blocking behavior of various empirical networks depend on the choice of the output nodes? From Eq. (4), the \mathcal{H}_2 -norm squared apparently depends on C . The authors should carefully address this issue.

Answer to Reviewer 2.2:

We thank the reviewer for their comment. The Reviewer is correct in stating that we didn’t consider specific choice of the output nodes in our study. This was done because we couldn’t find data that identified the output nodes in empirical networks. In fact, an extensive analysis of available datasets has shown that input nodes are sometimes identified but information on the output nodes is rarely provided. In the absence of the data, we just could not perform a comparison of different networks in terms of the specific selection of their output nodes, as we have done with the input nodes. Although we agree with the Reviewer that this is unfortunate, we would like to point out that one of the main results of our paper is that the \mathcal{H}_2 -norm squared can be written as a summation of individual contributions from each one of the output nodes (see Sec. D of the Methods and in particular Eq. (35)), so our choice of picking $C = I$ is equivalent to considering all these contributions, and any other choice of the matrix C would correspond to picking a subset of the data we have considered.

Comment Reviewer 2.3:

Higher-order interactions? Higher-order interactions are ubiquitous in most biological and ecological networks (food webs, connectomes, gene regulatory networks, KEGG pathways), because the underlying mechanisms are combinatorial and context-dependent. The authors should discuss if the presence of high-order interactions will affect their LTI modeling framework and their findings.

Answer to Reviewer 2.3:

We thank the reviewer for this comment. In this work, the systems we considered involved pair-wise interactions. We focused on the linear dynamics around a stable fixed point, so our analysis is restricted to the first-order (Jacobian) approximation. Effective higher-order interactions may emerge for phase-reduced periodic dynamical systems (such as many biological systems) when one goes beyond the first order [9], so that considering pair-wise interactions still represents the dominant part of the response. From this perspective, many-body interactions can, at leading order, be represented by an effective directed network of pair-wise couplings encoded in the Jacobian, as discussed in Ref. [10]. We have added a clarification to this effect in Sec. II A of the revised manuscript and now note in the Conclusions that the role of higher-order interactions remains an open question and a natural direction for future work.

Reviewer 4

Reviewer 4 evaluation:

The paper addresses an important and timely question: how complex networks process incoming signals. It seeks to distinguish between networks that amplify inputs - “passing networks” - and those that attenuate them - “blocking networks”. Using a control engineering perspective, the authors identify the structural properties that drive each behavior and then extend their analysis to a wide range of real-world networks, classifying them accordingly. Their main result is that directed acyclic graphs (DAGs) tend to facilitate passing, whereas symmetric and loopy networks are more effective at blocking. This, they suggest, may explain why natural systems often feature DAG structures, seeking to effectively transmit information, while engineered systems frequently favor architectures that, according to the paper’s analysis, suppress unwanted signal propagation.

Before turning to the detailed report, let me first offer my overall assessment of this submission.

The paper has several strengths:

1. The problem addressed is interesting and relevant.
2. The introduction, motivation and abstract are clear and suitably written for a broad readership.
3. Mathematical analysis is sound.
4. The results span a diverse set of networks from multiple domains.

The main points for improvement are:

1. While mathematically solid, the analysis feels narrower than the storyline suggests.
2. The exposition remains largely within an engineering framework. It could (and should) move beyond this to offer more "live" network dynamical insight.
3. At times the analysis is difficult to follow and not fully suited to an interdisciplinary audience.

Overall, I would be happy to see this work resubmitted after revision. Below I outline how I suggest to bring it into better shape for Nat. Comm.

Answer to Reviewer 4 evaluation:

We thank the Reviewer for their thorough reading of our paper and for their constructive comments. We are indebted to all Reviewers for their relevant contributions. The paper has been revised based on the feedback received. All the changes have been highlighted in blue.

Comment Reviewer 4.1:

Claimed generality. The study of information flow in complex networks has been rigorously advanced through a variety of mathematical approaches. Here, the authors adopt linear time-invariant (LTI) dynamics of the form given in Eq. (1). While this framework represents a central paradigm in control engineering, the paper should more clearly acknowledge its limitations in the broader study of network dynamics. Most relevant systems in this context, ranging from gene regulation to ecological interactions, are all inherently nonlinear. This is not a marginal issue: nonlinearity can fundamentally alter a network’s response patterns.

The authors seek to address this limitation by noting that nonlinear systems can be locally linearized around their fixed points. While correct, this does not fully resolve the problem. Distinct nonlinear dynamics defined on the same network, or even two different fixed points of the same dynamics, can lead to very different behaviors. Yet Eq. (1) remains unchanged. In

effect, the LTI formalism captures only the contribution of network structure, while overlooking the role of the system's intrinsic dynamics. However, modifying these underlying dynamics, even on a fixed structure, will change the effective weights A_{ij} and yield qualitatively different response patterns. It is therefore unsurprising that the authors' conclusions — e.g., DAGs as passing networks and symmetric ones as blocking — speak only to the effect of structure, without shedding light on the impact of nonlinear dynamics themselves

Answer to Reviewer 4.1:

We thank the Reviewer for this comment. The paper, methods, and supplementary information have undergone a major revision in order to address this important point. We fully agree that real-world systems such as gene regulatory networks and ecological communities are inherently nonlinear, and that nonlinearity can strongly influence how a network responds to perturbations. In the revised manuscript, we now make it clear that our analysis is based on the Jacobian of the underlying nonlinear dynamics, linearized at a fixed point, rather than on the raw adjacency matrix. The Jacobian keeps the same pattern of possible interactions as the structural network but assigns signed and weighted links that reflect activation, inhibition, excitation, damping, and other dynamical effects. A clear example of this has been added to the Methods, in the new section A. For each family of empirical networks, this Jacobian is derived from an appropriate nonlinear model (generalized Lotka–Volterra for food webs, Kuramoto oscillators for connectomes, and Michaelis–Menten–type models for pathways and genetic networks), so that the matrix used in the LTI framework reflects both the structure and the choice of underlying dynamics. Section E of the Methods introduces each one of the nonlinear models considered, the calculation of the fixed points, linearization, etc.

At the same time, we now explicitly acknowledge that the LTI description captures only the first-order (linear) contribution of the nonlinear system around a given fixed point. Different nonlinear dynamics on the same network, or different equilibria of the same model, can indeed produce different Jacobians and therefore different response patterns. This is a real limitation of any linearization-based approach. We now state this clearly in the Introduction and in the Conclusions. We also note, however, that linear analysis remains useful in important situations—such as near criticality or second-order phase transitions—where weak nonlinear effects are well approximated by the linearization and where the Jacobian gives meaningful insight into local signal propagation. Outside these circumstances, the linear approximation is naturally limited, and fully nonlinear analyses would be needed.

We have revised the manuscript accordingly, clarified these conceptual limitations, and updated Figs. 3 and 4 to reflect the results obtained using the Jacobian matrices of the real networks.

Comment Reviewer 4.2:

Results and exposition. The analysis relies on two closely related characterizations of system response: the H2 norm and the Bode diagram. While standard in control engineering, these measures may not align with how the network science community, or the domain-specific readers, intuitively conceptualize a network's response. I therefore encourage the authors to complement their current approach with perspectives that feel more natural to network researchers. For instance: What is the average magnitude of the network's response? How is it distributed across nodes? Does it depend on the degrees of the input and output nodes? This is not to suggest that the current mathematical measures are unsuitable. Just that they are too narrow, and should be enriched and reinforced by natural network-driven measures that can highlight dimensions of the problem, which, I believe, the readers will be curious about.

On this note, reading the paper, one senses a strong disconnect from the breadth of the story, as comes up from the introduction, to the actual results, which remain abstract and - at times - may seem naïve. For example, the paper, in principle, claims that by mapping the system's response to different frequencies, one can extract its response to all general types of signals. The argument rests on the fact that complex input signals can be decomposed into their Fourier series, thus breaking them down into a sum of simpler single frequency inputs. This is clearly correct, but remains as an abstract mathematical argument. The exposition would be truly empowered if the authors go ahead and show it. For instance, simulate actual responses of real-networks, to realistic signals. Make it "come to life". Show how a complex multi-frequency

input is stripped of off its high frequencies, as per the Bode diagram prediction. Demonstrate how you can use the paper’s theoretical insights to generate signals that pass vs. ones that are blocked. Illustrate visually how distinct networks process similar signals - as per the paper’s main prediction. As with most theoretical studies, there are two layers at play: the broad story told in the introduction and discussion, and the mathematical analysis that forms the technical core. The gap between them is bridged only through interpretation. To make that interpretation compelling, however, the figures and results must actively show the phenomena, not merely assert them. Biologists, ecologists, neuroscientists — and even physicists and engineers — may accept the mathematics, but they will only fully embrace the message once they can clearly "see" the outcomes brought to life.

Answer to Reviewer 4.2:

We thank the Reviewer for this comment. Based on this feedback, we have gone ahead and extensively revised the paper, methods and supplementary information. In particular, the new Figure 7 (also included here as Figure 2) describes a realistic example, based on a real connectome network and specific choices of the input and output nodes, to show how different input frequencies are either ‘passed’ or ‘blocked’ in the output signal. This provides a quantitative example that demonstrates the effect of signal transformation when passing through the network, in addition to the more abstract picture provided in Figure 1 of the paper.

Figure 2 shows an input signal that is composed of two sinusoidal signals with two different frequencies and two different amplitudes. The input signal enters the network through the blue node and results in the output signal through the red node. The output signal has the same two frequencies as the input signal, but with different amplitudes and phases. We see that the amplitude of the low-frequency input is slightly amplified (from $1 \rightarrow 1.41$), while the amplitude of the high-frequency input is damped (from $0.2 \rightarrow 0.002$).

Technical details:

The pair (A, B) is selected from our dataset for the connectome network “Macaque 30”. The network has 30 nodes and has 7 input nodes. Node 4 was selected as the input (shown as blue in Fig. 2) and node 1 was selected as the output (shown as red in Fig. 2). The distance from the input to the output node is $d = 2$. The input and the output signals are composed of the following components:

$$\text{Low Frequency input} = \sin(2\pi t), \quad (8a)$$

$$\text{High Frequency input} = 0.2 \sin(40\pi t), \quad (8b)$$

$$\text{Low Frequency output} = A_1 \sin(2\pi t + \phi_1), \quad (8c)$$

$$\text{High Frequency output} = A_2 \sin(40\pi t + \phi_2), \quad (8d)$$

where

$$A_1 = |G(2\pi J)| = 1.41, \quad \phi_1 = \angle G(2\pi J) = -86^\circ, \quad (9a)$$

$$A_2 = |G(40\pi J)| = 0.002, \quad \phi_2 = \angle G(40\pi J) = -251^\circ, \quad (9b)$$

where $J = \sqrt{-1}$. We see that the amplitude and the phases of the output components can be seen in the Bode plot in the bottom panel of Fig. 2 (shown as green circles for the low-frequency signal and purple squares for the high-frequency signal). Note that magnitudes of 1.41 and 0.002 correspond to dB values $20 \log_{10}(1.41) = 3$ and $20 \log_{10}(0.002) = -54$, respectively.

Comment Reviewer 4.3:

Analysis. The analysis is highly “engineery” in style. This is not a problem in itself, but it does present challenges for the intended readership, many of whom are not control engineers. Even I, while not an engineer but well rooted in the network dynamics domain, found myself needing to look up certain concepts and struggling with some of the mathematical transitions. For example, Eq. (2) is not immediately intuitive, and Eq. (3) introduces the $20 \log_{10}$ factor without explanation or reference to the Methods or a relevant citation (later understood to represent decibel units). The same issue arises with the definitions of the H2 norm and the Gramian in Eqs. (4)–(6), which are presented without justification or intuitive grounding. Similarly, Eqs. (9) and (10) appear imposed rather than derived, again without explanation. To be clear, it is

FIG. 2: The top panel shows the passing behavior of a low-frequency input signal and blocking a high-frequency one. The bottom panel shows the Bode plot associated with the system.

perfectly acceptable to relegate detailed derivations to the SI or Methods section — the reader does not need to follow every technical step. But the exposition should not force readers to decode what feel like unexplained or “magical” transitions. At minimum, the authors should provide references, intuitive explanations, or brief contextual notes that make the mathematical steps accessible and allow readers to grasp the underlying principles that enable them.

Answer to Reviewer 4.3:

We thank the reviewer for this helpful comment. We have rewritten the first part of Sec. II to provide a clearer and possibly smoother transition between concepts. All the changes have been highlighted in red in the revised manuscript. We added references to the classical textbooks by Ogata and Kailath to provide the needed background to readers who are not experts in control theory. We believe the textbook by Ogata is a better reference for concepts introduced in the frequency domain and the textbook by Kailath is a better reference for concepts introduced in the time domain. Below we provide detailed responses to the individual points raised by the Reviewer.

- This sentence has been added right after Eq. (3): "The factor $20 \log_{10}(\cdot)$ expresses the magnitude $|G(j\omega)|$ in decibels, the conventional logarithmic unit for system gain."
- Extensive text has been added after Eqs. (4) and (5) to explain them.
- A reference to the classical textbook by Kailath has been added for the Lyapunov equation (Eq. (6))
- An explanation has been added right after Eq. (7), which leads into Eq. (8)
- Additional text has been added right after Eq. (9) to provide additional information.
- Additional text has been added right after Eq. (10) to provide additional information.

We hope the Reviewer will find our work satisfactory.

Comment Reviewer 4.4:

A further consequence of the paper’s opaque derivations is that it is not always clear which results are canonical and which represent the paper’s original contributions. Overall, the mathematics primarily involves linear algebraic manipulations grounded in well-known principles. However, the absence of references or explanatory notes makes it difficult at times to discern which steps are established and which are newly introduced in this work.

Answer to Reviewer 4.4:

We thank the Reviewer for this comment. We certainly did not intend for any part of our work to appear opaque, and we appreciate the opportunity to clarify and improve the presentation. In light of this feedback (and as described in our response to Comment 4.3), we have revised the manuscript to make the mathematical derivations and their provenance more transparent. Moreover, we now have added a list of results that constitute new contributions of this work in the Conclusions. Section IIB contains most of our new work, while the “Methods” Section provides the corresponding derivations and supporting explanations, which is mostly new as well. We hope these revisions make the distinction between established material and novel contributions clear. The new contributions include: (i) our analysis of the frequency response for an arbitrary network and our sketch of the Bode plots that depend on the network topology and the particular selection of the input and output nodes, (ii) the study of the \mathcal{H}_2 norm for several empirical networks, (iii) our derivations for the case of directed acyclic graphs, (iv) our derivations for the DC gain, (v) our approximate formula for the trace of the Gramian in the case in which the largest real part of the eigenvalues of the matrix A is large in magnitude and (vi) the modularity property for the \mathcal{H}_2 -norm squared (to the best of our knowledge, the modularity property was known with respect to the input nodes but not with respect to the output nodes.) As already stated, the additional sentences added at the end of the first paragraph of the Conclusions summarize the main new contributions of the paper.

Comment Reviewer 4.5:

Figures. Figure 2 looks more like an illustration than an actual measurement. Axes should be labeled clearly - especially log vs. linear scale. It took me some time to understand that the slope of $-20N$ represents $1/w^N$...

Answer to Reviewer 4.5:

We thank the reviewer for this comment. Figure 2 has now been updated by the addition of gridlines to emphasize the log vs linear scale.

Comment Reviewer 4.6:

Table 1 is supposed to convince us that H_2 decreases with c in a way that dominates its growth with d . I assume that this is rooted in the higher exponent applied to c . Would not it be more natural to show this as a figure, rather than a Table?

Answer to Reviewer 4.6:

We thank the reviewer for their comment. The discussion that accompanies Table 1 has been modified in the revised manuscript—all changes highlighted in color. The purpose of the table is to show that the coefficients that multiply the powers of c and \bar{w} decay ‘slowly’ with the number of nodes N , i.e., in a sublinear way. We think that reporting the numbers in a table, as well as the analytical formula, is a good way to show this. Because of the slow decay of the coefficients, whether the \mathcal{H}_2 norm increases or decreases with N is determined by the balance between the self suppression coefficient c and the node-to-node average stimulation coefficient $|\bar{w}|$. This has been now clarified and commented on in the revised manuscript. We have now added a proof for our formula, see Supplementary Material Section 1A. Moreover, we have derived an additional formula for the case in which the self-coupling loops at different nodes along the chain are different, see Supplementary Material Section 1B. The Reviewer will find all this additional

work in the revised version of the manuscript.

Comment Reviewer 4.7:

I feel that Figure 3a would be more informative if the x-axis would be presented in log-scale. Many of the symbols consolidate around small p , but as p -values go, it often matters if p 10^{-5} or 10^{-8} - a distinction that is fully obscured in linear scale.

Answer to Reviewer 4.7:

We thank the reviewer for this comment. We tried to change the scale on the x-axis of Fig. 3a to be logarithmic but unfortunately, we found that this modification did not improve the readability of the figure and so decided to leave it as is.

Comment Reviewer 4.8:

Minor typos. Fig. 3 caption "the numbers of randomly input". I assume you meant "random". Text below Figure 2 " w_1, \dots, w_N ". I think it should be w_{N-1} .

Answer to Reviewer 4.8:

We thank the reviewer pointing out these errors and helping us improving the manuscript clarity. The Reviewer is correct for both points. In the caption of Figure 3, we have corrected "the numbers of randomly input" to "The number of randomly chosen input nodes". For the unidirectional chain network in Figure 2, we have corrected the text to refer to the parameters as " w_1, w_1, \dots, w_{N-1} "

Comment Reviewer 4.9:

Conclusion. As noted, the paper addresses an interesting problem and the underlying mathematics is rigorous. However, the presentation is overly narrow and formal. I strongly encourage the authors to move beyond the abstract formalism and explicitly demonstrate the broader implications and interpretations that make their results relevant to the interdisciplinary network dynamics community. It is also crucial that the paper clearly acknowledge the limitations of the LTI framework and its potentially restrictive applicability to real-world, nonlinear systems.

Answer to Reviewer 4.9:

We agree that they are highly relevant and complement our work well. We also acknowledge that the original version of the manuscript did not sufficiently emphasize how our contribution fits within the broader literature. To address this, and as also outlined in our response to Reviewer 1, we have substantially improved the presentation and narrative of the paper. In particular, we have smoothed several aspects of the mathematical formalism to improve readability (for example, by simplifying the definitions of the matrices B and C), removed terminology that was not essential for our purposes (such as "controllability" associated with the Gramian or references to Metzler matrices), and clarified concepts such as the use of z -scores and p -values for a broader audience. We have also strengthened the narrative flow by summarizing the main conclusions of each section and by more clearly anticipating the material that follows, ensuring smoother transitions between parts of the manuscript. Finally, we now explicitly acknowledge several limitations of the present work—such as the neglect of nonlinear terms, higher-order interactions, and time-dependent Jacobians—and outline these as natural directions for future research. We are confident that these revisions improve both the clarity and accessibility of the paper.

Comment Reviewer 4.10:

A final note on relevant literature. I do not want to seem like I am "fishing" for citations, but I do want to refer the authors to some relevant papers that I co-authored. I feel slightly uncomfortable doing so, however - in this case - I really believe that the authors will find them relevant, and that they can help situate the current manuscript within a broader context. I also wish to assure the authors that my assessment in round 2 of this paper, will be unaffected by their choice to cite or ignore these papers. I am truly attempting to be constructive here, not to enforce a reference :-) Over the past several years, my co-authors and I have focused on the mathematical characterization of nonlinear network dynamics - specifically, we analyzed the timescales and Jacobian structure of the system's linear response around its fixed-points. Hence, our work is intricately related to the story of the current paper. Let me mention 2 recent publications that, I think, are highly relevant and can contribute to the current paper's narrative:

1. Spatiotemporal signal propagation in complex networks. *Nature Physics* **15**, 403–412 (2019). We consider the system's response to DC-like signals. We find that for a given (scale-free) network, the nonlinear dynamics impose distinct universality classes of propagation patterns. These universal patterns can be characterized by a single predictable exponent (θ) that predicts the intrinsic response timescales of the hub-nodes.

2. Emergent stability in complex network dynamics. *Nature Physics* **19**, 1033–1042 (2023). We derive the Jacobian of nonlinear network dynamical systems, and find how its weights change under different nonlinear models. We show that they do not change slightly, but rather very dramatically as one transitions between nonlinear models. This touches on the relevance of the LTI framework used in the current work. For the LTI to truly approximate the nonlinear response around a fixed-point, as stated in the manuscript, one must use the appropriate Jacobian weights.

In my understanding a system's response to external signals is mediated by both the network structure and the nonlinear dynamics. The current submission primarily addresses the former, while the papers I mention above complement the analysis by elucidating how nonlinear dynamics shape response patterns. As I indicated, I do not intend to count citations in the revision round. I just think these papers can help better place the current contribution in its broader context.

I wish the authors the best of luck and look forward to reading the revised submission,
Baruch Barzel

Answer to Reviewer 4.10:

We thank the reviewer for their comment.

We thank the Reviewer for providing these two references. We agree that they are relevant to our work and nicely complement it. We added a discussion in the introduction to emphasize the contribution of our work, that is, whether various technological and natural systems with specific network structure and dynamics, tend to block or to facilitate signal propagation. In that sense, our work accounts for both the network topology and the dynamics. We added a note about this in the introduction to clarify the contribution of our work.

-
- [1] M. Asllani, R. Lambiotte, and T. Carletti, *Science advances* **4**, eaau9403 (2018).
 - [2] J. D. O'Brien, K. A. Oliveira, J. P. Gleeson, and M. Asllani, *Physical Review Research* **3**, 023117 (2021).
 - [3] R. Nartallo-Kaluarachchi, M. Asllani, G. Deco, M. L. Kringelbach, A. Goriely, and R. Lambiotte, *Phys. Rev. E* **110**, 034313 (2024).
 - [4] R. M. May, *Nature* **238**, 413 (1972).
 - [5] S. Allesina and S. Tang, *Nature* **483**, 205 (2012).
 - [6] K. Z. Coyte, J. Schluter, and K. R. Foster, *Science* **350**, 663 (2015).
 - [7] G. Karlebach and R. Shamir, *Nature reviews Molecular cell biology* **9**, 770 (2008).
 - [8] C. Meena, C. Hens, S. Acharyya, S. Haber, S. Boccaletti, and B. Barzel, *Nature Physics* **19**, 1033 (2023).
 - [9] I. León and D. Pazó, *Physical Review E* **100**, 012211 (2019).
 - [10] F. Della Rossa, D. Liuzza, F. Lo Iudice, and P. De Lellis, *Physical Review Letters* **131**, 207401 (2023).

We thank the Editor for their careful handling of our paper and all Reviewers for their thorough reading of our paper and for their constructive comments. We provide detailed responses below.

Reviewer 1

Reviewer 1 evaluation:

Nazerian et al. have thoroughly revised their manuscript to address the multitude of comments and requests by three Reviewers. In particular, they have reasonably addressed the points I have raised, better explained the technical details and complemented them with intuitive explanations and discussion where appropriate. I list below a few minor points that can still be improved.

Answer to Reviewer 1 evaluation:

We thank the Reviewer for the additional feedback provided. The paper has undergone a minor revision based on the feedback received. All the changes have been highlighted in blue.

Comment Reviewer 1.2:

In addition, I would like to note that I found two points raised by the two other Reviewers of particular importance. First, comment 1 by Reviewer 2 explicitly raised the question about the direct relation between the graph's weight matrix and the dynamical system properties resulting in the matrix elements A_{ij} , as well as the relevance of this relation. I think the authors have reasonably addressed this issue within the scope of the current work. I particularly value the new section E of the Appendix/Supplement that helps orient readers. It might still make sense to discuss in the outlook any limitations for practical applicability that might result from the mathematical conditions (non-negativity) required.

Answer to Reviewer 1.2:

We very much appreciate the feedback received by the Reviewer. We have added a sentence to the Conclusions about the limitations that come from the non-negativity condition.

Comment Reviewer 1.3:

Second, Reviewer comment 4.2. (along with 4.1) might be particularly valuable for connecting the general mathematical results to the broad interdisciplinary readership of Nature Communications. I believe showcasing specific examples, as attempted in the revised version of the manuscript, may make the work more accessible. minor comments:

- The abstract is lengthy, perhaps interactions with the Editors might help further improve its reach
- non-normality (with the "-") on page 1
- I like the explicit reference on page 3 to the Methods section where A and A -tilde are explicitly related.
- I think the direct connection "bringing the results to life" (Fig. 7) should be presented and discussed within main manuscript, perhaps complemented by additional illustration/s in the main part of the work, I leave the Editors to judge this issue.
- there are a few layout/setting issues in the reference list, e.g. empty parentheses "()".

Answer to Reviewer 1.3:

We very much thank the Reviewer for their feedback. The abstract has been fully rewritten to comply with the journal's word limits, the reference list has been revised, and the typographical issue concerning "non normality" has

been corrected. Regarding the previous comments 4.1 and 4.2, we note that these points were addressed in a previous revision by expanding the discussion of concrete examples to better connect the general mathematical results to a broad interdisciplinary readership. In the present version, we therefore did not introduce additional examples or move Fig. 7 to the main text, as we believe this would be repetitive with existing material, given that Fig. 7 is qualitatively very similar to Fig. 1, and would not add new conceptual insight.

Reviewer 2

Reviewer 2 evaluation:

The authors have successfully addressed all my previous concerns.

Answer to Reviewer 2 evaluation:

We thank Reviewer 2 for their positive assessment of our work and for all the extremely relevant feedback they have provided.

Comment Reviewer 2.1:

One minor issue: A plus sign is missing in Eq.(1) of the response letter (corresponding to Eq.(36) of the main text).

Answer to Reviewer 2.1:

We thank the Reviewer for pointing this important typo out. We have now corrected it in the final version of the paper.

Reviewer 4

Reviewer 4 evaluation:

I have now reviewed the revised submission and believe the authors have done an excellent job elevating the paper beyond a narrow control-engineering perspective. I particularly appreciate the added analysis of nonlinear systems using real-world Jacobians, which provides a substantial and meaningful expansion of the paper's theoretical scope. Overall, the current version is now strengthened, and I recommend publication without further changes.

Answer to Reviewer 4 evaluation:

We thank Reviewer 4 for their positive assessment of our work and for all the extremely relevant feedback they have provided.